# Single-cell RNA sequencing reveals the fragility of male spermatogenic cells to Zika virus-induced complement activation

Wei Yang[1,4], Li-Bo Liu[1,4], Feng-Liang Liu ®[2,4], Yan-Hua Wu[1,4], Zi-Da Zhen ®[1], Dong-Ying Fan[1], Zi-Yang Sheng[1], Zheng-Ran Song[1], Jia-Tong Chang[1], Yong-Tang Zheng ®[2] ✉, Jing An ®[1,3] ✉ & Pei-Gang Wang ®[1] ✉

Zika virus (ZIKV) is a potential threat to male reproductive health but the mechanisms underlying its influence on testes during ZIKV infection remain obscure. To address this question, we perform single-cell RNA sequencing using testes from ZIKV-infected mice. The results reveal the fragility of spermatogenic cells, especially spermatogonia, to ZIKV infection and show that the genes of the complement system are significantly upregulated mainly in infiltrated S100A4 + monocytes/macrophages. Complement activation and its contribution to testicular damage are validated by ELISA, RT–qPCR and IFA and further verify in ZIKV-infected northern pigtailed macaques by RNA genome sequencing and IFA, suggesting that this might be the common response to ZIKV infection in primates. On this basis, we test the complement inhibitor C1INH and S100A4 inhibitors sulindac and niclosamide for their effects on testis protection. C1INH alleviates the pathological change in the testis but deteriorates ZIKV infection in general. In contrast, niclosamide effectively reduces S100A4 + monocyte/macrophage infiltration, inhibits complement activation, alleviates testicular damage, and rescues the fertility of male mice from ZIKV infection. This discovery therefore encourages male reproductive health protection during the next ZIKV epidemic.

Zika virus (ZIKV) belongs to vector-borne viruses, but unlike other members in the family, it can cause infection in testes and result in relevant symptoms such as oligospermia and hematospermia, making ZIKV a potential threat to male reproductive health[1,2]. Although many cells including Sertoli cells, spermatogenic cells, and macrophages in mice and nonhuman primates have been proven to be susceptible to ZIKV infection[3], the influence of ZIKV infection on spermatogenic cells and the underlying mechanisms remain to be elucidated.

Spermatogenic cells normally reside inside the seminiferous tubules, which are the basic unit of the testis and provide a suitable environment for spermatogenesis. A blood-testis barrier (BTB), mainly formed by the seminiferous epithelium (Sertoli cells) and tight junctions, is located near the base of the seminiferous tubule and divides the epithelium into basal and abluminal compartment. Spermatogonia reside in the basal compartment and are insulated from circulating immune cells only by the basal membrane, whereas spermatocytes and spermatids reside in the abluminal compartments. Thus, the BTB sequesters germ cells residing in the abluminal compartment from the immune system to provide an immune-privileged microenvironment for the development of spermatogonia to spermatids[4].

[1]Department of Microbiology, School of Basic Medical Sciences, Capital Medical University, Beijing 100069, China. [2]Key Laboratory of Animal Models and Human Disease Mechanisms of Chinese Academy of Sciences, Kunming Institute of Zoology, Chinese Academy of Sciences, Kunming, Yunnan 650107, China. [3]Center of Epilepsy, Beijing Institute for Brain Disorders, Beijing 100093, China. [4]These authors contributed equally: Wei Yang, Li-Bo Liu, Feng-Liang Liu, Yan-Hua Wu. ✉e-mail: zhengyt@mail.kiz.ac.cn; anjing@ccmu.edu.cn; pgwang@ccmu.edu.cn

Both spermatogenic cells and Sertoli cells are susceptible to ZIKV infection, however, human spermatogenic cells have been demonstrated to support ZIKV replication in vitro without obvious changes in cell viability[3], suggesting that ZIKV infection itself is not the main cause of spermatogenic cell damage. Recently, we demonstrated that myeloid S100A4+ macrophages are recruited into ZIKV-infected testes, secrete IFN-γ to increase the permeability of the BTB, and assist ZIKV invasion and persistence in seminiferous tubules[5]. We hypothesized that S100A4+ macrophages participated in spermatogenic cell damage, either through direct endocytosis or by facilitating CD8+ T cells to access ZIKV-infected spermatogenic cells. However, not as expected, we found that S100A4+ macrophages did not directly eradicate ZIKV-infected spermatogenic cells, and CD8 T cells could not access the intraluminal space and not interact with S100A4+ macrophages. Moreover, instead of slowly disappearing, all spermatogenic cells in a seminiferous tubule diminished very quickly, implying that some other mechanisms are involved.

Here, we show that the classical activation of the complement system is the main reason for the damage to spermatogenic cells in ZIKV-infected testes. C1q, a key molecule of the classical pathway, is expressed and released into seminiferous tubules by S100A4+ monocytes/macrophages, and then is activated to form a membrane-attacking complex (MAC) on ZIKV-infected spermatogenic cells. On this basis, we further determine that niclosamide, an S100A4 inhibitor, can alleviate ZIKV-induced testicular damage and protect the fertility of male mice.

## Results

### Cell clusters in ZIKV-infected mouse testis defined by scRNA-Seq

To investigate the influence of ZIKV infection on testes, testicular cells from ZIKV-infected (14 dpi.) and uninfected A6 male mice (*Ifnar*$^{-/-}$ mice) were analyzed by single-cell RNA sequencing (scRNA-Seq). After filtering out poor-quality cells, 11014 cells in control testes and 11974 cells in ZIKV-infected testes were identified as 9 cell clusters (Fig. 1a and Supplementary Fig. S1a), based on the shared nearest neighbor (SNN) module optimization algorithm and the expression of known cell type-specific markers[6]. Of them, Leydig cells, spermatogenic cells, Sertoli cells (suspected), and fibroblasts were testicular resident cells and were present in both samples (Fig. 1a). Notably, the number of spermatogenic cells was greatly reduced after ZIKV infection (Fig. 1a and Supplementary Fig. S1b–d). In contrast, monocytes, macrophages, granulocytes, fibroblasts, and T lymphocytes dramatically increased in ZIKV-infected testes (Fig. 1a, b and Supplementary Fig. S1b). Similar to previous reports[7], Sertoli cells were difficult to identify by scRNA-seq so they were labeled as Sertoli (suspected), and the testicular macrophages were too few to be distinguished from other macrophages after ZIKV infection, therefore, these cells were not separately analyzed in the following research.

In ZIKV-infected testes, ZIKV RNA was detected in most cells excluding Leydig cells and fibroblasts (Fig. 1c), suggesting their susceptibility to ZIKV infection. These results confirmed previous reports by our group and other groups that various cell types in the testis can be infected by ZIKV[5]. Intriguingly, Leydig cells and fibroblasts were the only two testicular resident cell types whose number increased after ZIKV infection, suggesting a solid association between ZIKV infection and damage to testicular resident cells.

To evaluate the contribution or impact of each type of cell in the microenvironment of ZIKV-infected testes, the expression of a panel of cytokines and their signaling were analyzed by the cell–cell communication analysis tool CellChat[8]. Compared to uninfected testes, 27 signaling pathways were significantly activated in ZIKV-infected testes (Fig. 1d). A heatmap of the relative importance of cell clusters in the activation of each signaling pathways showed that monocytes and macrophages ranked at the top in both incoming and outgoing signaling patterns (Fig. 1d and Supplementary Fig. S1e), suggesting that

they had the greatest contribution and impact on the microenvironment of ZIKV-infected testes. In contrast, the contribution and impact of spermatogenic cells and Sertoli cells, rather than Leydig cells, to the activation of these signaling pathways were much weaker (Fig. 1d and Supplementary Fig. S1e). One possible explanation was that cells in the seminiferous tubule were partially insulated from the inflammatory environment by the BTB. Of the cytokines and factors, CCL, MIF and complements ranked at the top (Fig. 1d), implying that they played important roles in ZIKV-infected testes. The complement system is particularly notable, because it can kill target cells by forming a MAC in the membrane.

### Characterization of ZIKV-infected spermatogenic cell clusters

The initial clustering analysis indicated that the spermatogenic cell cluster comprises several cell subsets, so we performed further clustering analysis only on spermatogenic cells. Based on markers reported recently[6,9], spermatogenic cell clusters were identified and clarified into spermatogonia, spermatocytes, and spermatids (Fig. 2a, b). DDX4, a marker of spermatogenic cells, was expressed by all subsets (Fig. 2c). Spermatogenic cells were also subjected to pseudotime analysis, and the results validated the differentiation from spermatogonia to spermatids in uninfected testes (Fig. 2d). Intriguingly, the mitosis of spermatogonia in ZIKV-infected testes was significantly activated, as the expression of S, G2-M, and S-G2-M phase genes (*Top2a*, *Cdk1*, and *Mki67*, respectively) was increased in these cells (Fig. 2e). Pseudotime analysis also showed that the differentiation of spermatogonia to spermatids was altered to another subcluster of spermatocytes in ZIKV-infected testes, in which the expression of *Mlh3*, *Hormad1* and *Sycp3*, three critical genes for meiosis[9], was reduced and consequently the differentiation to spermatids was inhibited (Fig. 2e).

ZIKV RNA was detected in all subsets with similar percentages, suggesting that they were all susceptible to ZIKV infection (Fig. 2f). Nevertheless, the expression level of ZIKV RNA decreased with the development from spermatogonia through spermatocytes to spermatids (Fig. 2f). Spermatogonia and early differentiated spermatocytes are thought to be the stem cells for spermatogenesis[10], and these results thus demonstrated a correlation between the susceptibility and the "stemness" of spermatogenic cells. Neural progenitor cells are well-known target cells for ZIKV in the brain[11,12]. Here we showed that spermatogenic stem cells were also greatly susceptible to ZIKV infection. The association between stemness and ZIKV infection is an interesting question and worthy of further investigation.

In comparison to spermatocytes, ZIKV-infected testes lost more spermatogonia and spermatids (Fig. 2g). The reduction in the number of spermatids possibly resulted from the alteration of spermatocyte differentiation induced by ZIKV infection (Fig. 2d, e), but the reasons for the loss of spermatogonia remain unknown. As most spermatocytes survived in the presence of ZIKV RNA, ZIKV infection itself was unlikely to be the cause of spermatogenic cell damage. To determine how spermatogenic cells were damaged, the expression of genes related to known death pathways was analyzed in spermatogenic cells. In general, the genes related to apoptosis, proptosis, necrosis, autophagy, and ferroptosis in spermatogenic cells looked similar between the control and ZIKV-infected testes, with a slight increase after ZIKV infection (Fig. 2h). These results were validated by IFA in ZIKV-infected testes (Supplementary Fig. S2a–d), in vitro cultured primary testicular cells of mice and the human seminoma cell line JKT-1 (Supplementary Fig. S2e, f). According to these results, as well as previous reports[3], ZIKV infection itself did not result in great damage to spermatogenic cells.

### Characterization of monocytes/macrophages in ZIKV-infected testes

To characterize immune cells infiltrated in ZIKV-infected testes, their gene profiles were analyzed by SingleR (Single-cell Recognition)[13]. Most lymphocytes in the T/NK cell cluster were identified to be CD8+ T

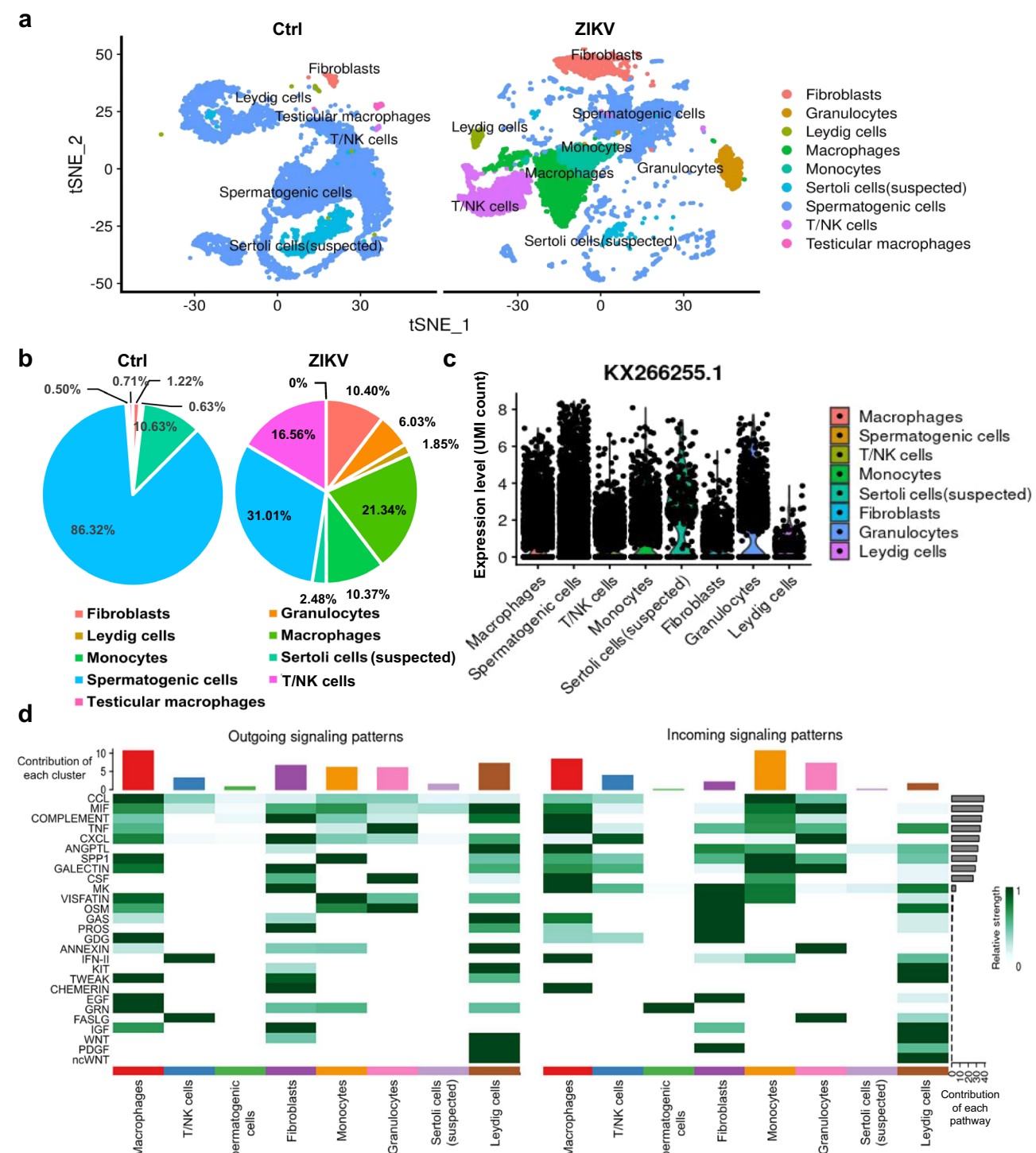

**Fig. 1 | Identification of cell clusters in ZIKV-infected testes.** Testes from PBS-injected (control) or ZIKV-infected A6 male mice at 14 dpi were subjected to single-cell sequencing analysis. **a** The distribution of cell clusters in the testes from control and ZIKV-infected mice was shown in tSNE chart. **b** The proportion of each cell cluster in all cells in the testes from control and ZIKV-infected mice was shown in pie chart. **c** The expression of ZIKV RNA in all cell clusters was shown in violin plot. **d** Intercellular communication in testes was analysis by using CellChat. The relative importance of each cell cluster based on the computed network centrality measures of outgoing/incoming signaling network was shown in heatmap. The relative contribution of each cluster or pathway was shown in corresponding column chart.

lymphocytes with high expression levels of granzyme B (GZMB) (Supplementary Fig. S3a). However, IFA rarely showed co-staining between GZMB and DDX4+ cells (Supplementary Fig. S3b), suggesting that the loss of spermatogenic cells was not attributed to cytosolic T cells.

Monocytes and macrophages were the dominant cell clusters in ZIKV-infected testes. As revealed by Kyoto Encyclopedia of Genes and Genomes (KEGG) enrichment analysis, macrophages had gene

expression similar to that of monocytes (Supplementary Fig. S4a, b). Gene set enrichment analysis (GSEA) further showed that the genes were related to phagocytic function and the inflammatory pathway such as Lysosome, Phagosome, TNF signaling pathway were similarly enriched in these two clusters (Supplementary Fig. S4c), indicating that the macrophages in ZIKV-infected testes were mainly differentiated from monocytes.

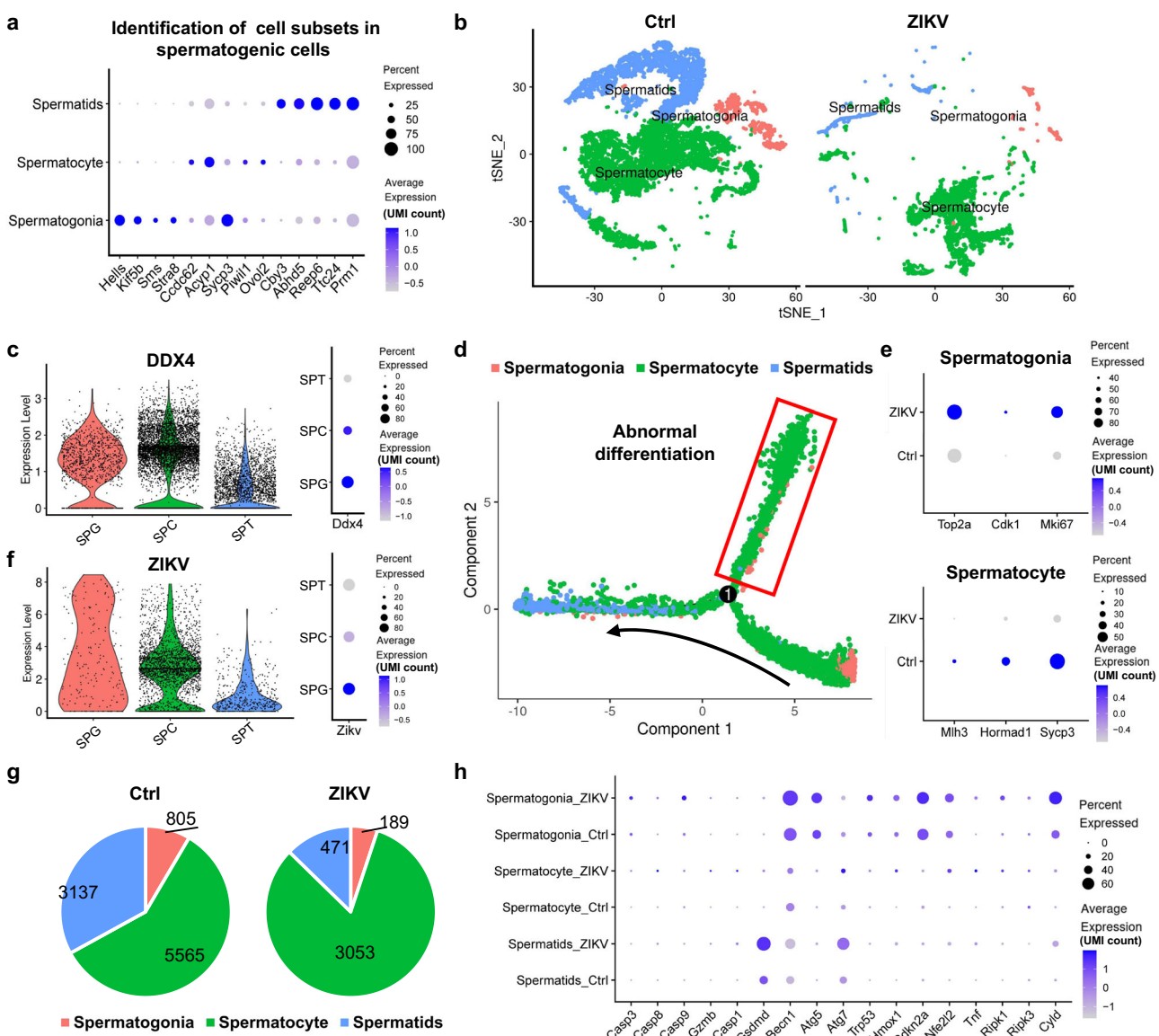

**Fig. 2 | Characterization of spermatogenic cells in control and ZIKV-infected testes by Single-cell sequencing. a** Spermatogenic cells in the testes from control and ZIKV-infected mice were sorted into spermatogonia, spermatocyte, and spermatids subsets. The expression of spermatogenic maturity-related genes were shown in bubble chart. **b** The distribution of spermatogonia, spermatocyte, and spermatids were shown in tSNE chart. **c** The expression of DDX4 gene in spermatogonia (SPG), spermatocyte (SPC), and spermatids (SPT) subsets were shown in violin plot and bubble chart. **d** Pseudotime analysis of spermatogonia, spermatocyte, and spermatids in the testes from control and ZIKV-infected mice. The curve with arrowhead showed the differentiation in control testes. Cells shown within red rectangular appeared mainly in ZIKV-infected testes. **e** Expression of mitosis-related genes in spermatogonia and meiosis-related genes in spermatocyte in the testes from control or ZIKV-infected testes was shown in bubble chart. **f** The expression of ZIKV RNA in spermatogonia (SPG), spermatocyte (SPC), and spermatids (SPT) subsets were shown in violin plot and bubble chart. **g** The number of spermatogonia, spermatocyte, and spermatids in the testes from control and ZIKV-infected mice. **h** Expression of genes related to apoptosis, pyroptosis, necrosis, autophagy, and ferroptosis in spermatogonia, spermatocyte, and spermatids in the testes from control and ZIKV-infected mice.

The differentiation of monocytes into macrophages was then subjected to pseudotime analysis, and the ZIKV RNA level was also analyzed during differentiation. As shown by pseudotime analysis, most monocytes differentiated into macrophages (shown as Mo/Ma differentiation) (Fig. 3a). Nevertheless, a fraction of monocytes turned into a state that was likely to be apoptotic (Fig. 3a), as revealed by the high level of apoptosis or pyroptosis genes and low level of monocyte marker genes (Fig. 3b, c and Supplementary Fig. S5a, b). ZIKV RNA was detected in both monocytes and macrophages, but its level was lower in apoptotic monocytes (Fig. 3c, d). The presence of ZIKV RNA and its change implied that monocytes might play an important role in the dissemination of ZIKV in the testis. Although apoptosis or pyroptosis of monocytes could inhibit ZIKV replication, most ZIKV evades these

processes and promotes the differentiation of monocytes to macrophages into the testis.

The top 20 upregulated genes and downregulated genes during the differentiation from monocytes to macrophages were analyzed. In line with the results from CellChat, the genes of the complement system, including *C1qa, C1qb,* and *C1qc*, were significantly upregulated (Supplementary Fig. S5c, d). Analysis of all cell clusters further indicated that other complement components, such as *C1ra, C1s1,* and *C3*, were also upregulated in monocytes, macrophages, or fibroblast-like clusters after ZIKV infection (Fig. 3f–h). The complement system can be activated by the classical pathway, alternative pathway, or mannose-binding lectin (MBL) pathway[14]. Of the three pathways, the expression levels of molecules related to the classical pathway

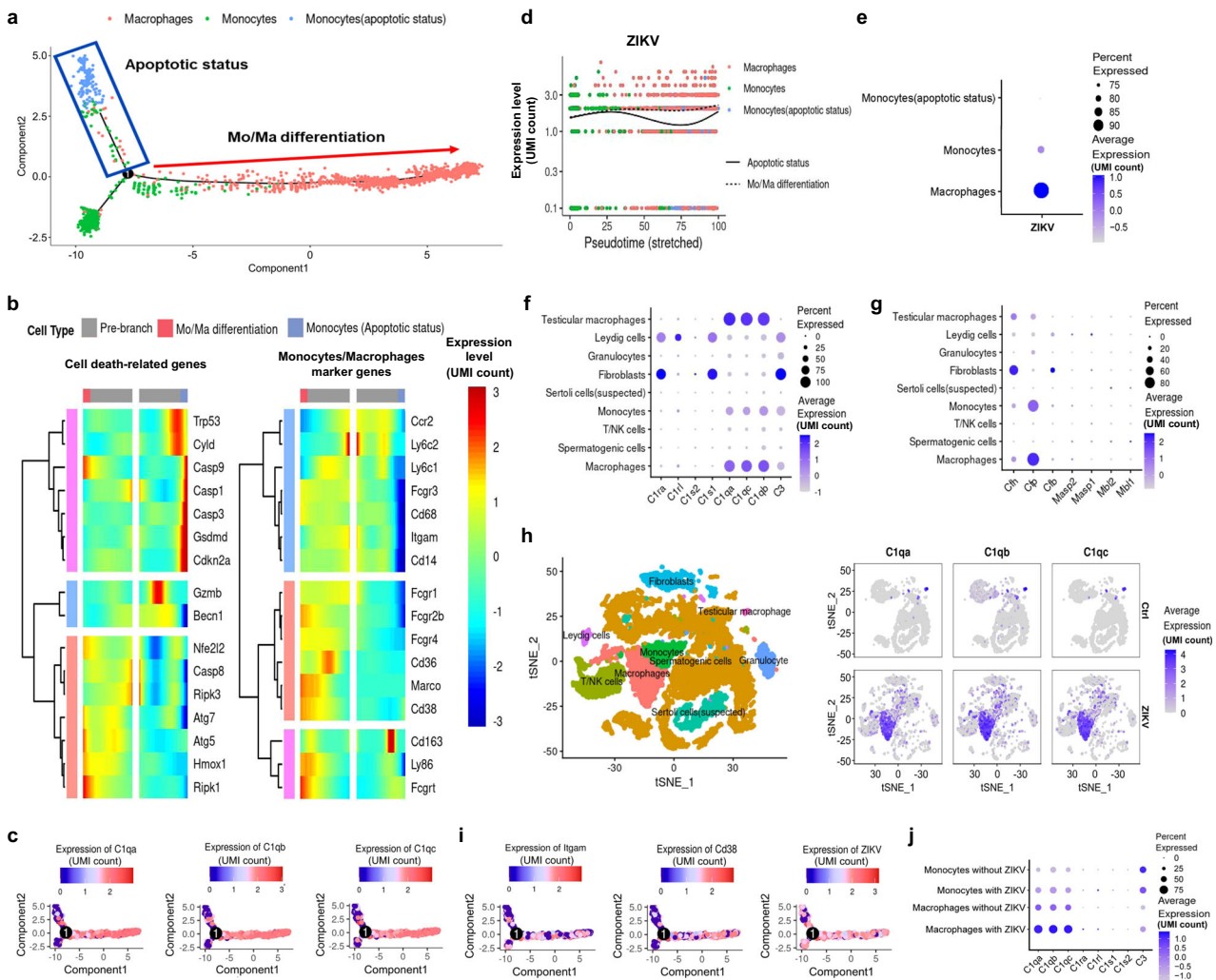

**Fig. 3 | Characterization of monocytes and macrophages in control and ZIKV-infected testes by Single-cell sequencing. a** Pseudotime analysis of monocytes and macrophages in the testes from ZIKV-infected mice. **b** The expression of cell death-related genes and monocytes/macrophages marker genes were shown in heatmap. **c** The signatures and expression dynamics of *Itgam* and *Cd38* and ZIKV in monocytes and macrophages were shown in pseudotime progression. **d, e** The expression of ZIKV RNA in monocytes and macrophages was shown in line chart (**d**) and bubble chart (**e**). **f, g** The expression of genes involved in classical complement activation pathway (**f**), and alternative/MBL complement activation pathway (**g**) in all cell clusters. **h** The expression of *C1qa, C1qb, C1qc* gene in cell clusters were shown in tSNE chart. **i** The signatures and expression dynamics of the expression of *C1qa-c* in monocytes and macrophages were shown in pseudotime progression. **j** The influence of ZIKV infection on the expression of complement genes in macrophages or monocytes was shown in bubble chart.

(*C1qa-c, C1ra, C1s1*) were all upregulated, especially in macrophages (Fig. 3f–i). The activating and inhibiting factors of the alternative pathway (*Cfp* and *Cfh*) were both upregulated (Supplementary Fig. S6a–e), and those of the MBL pathway (*Mbl1, 2, Masp1* and *2*) showed no change (Supplementary Fig. S6d, e). As most monocytes and macrophages have been infected with ZIKV (Fig. 3e), the influence of ZIKV infection on *C1q* expression was also analyzed. Compared with uninfected monocytes or macrophages, the expression levels of *C1qa-c* in macrophages were all enhanced upon ZIKV infection (Fig. 3j), further implying that they might play important roles in ZIKV-infected testes.

**Classical complement activation resulted in spermatogenic cell damage**

We then used RT–qPCR and ELISA to verify the results of scRNA-seq. In line with this finding, RT–qPCR results showed that the expression level of *C1q* was significantly increased (Supplementary Fig. S7a), while those molecules related to the activation of the MBL pathway or alternative pathway were not significantly upregulated (Supplementary Fig. S7b–e). ELISA displayed similar results as RT–qPCR. In ZIKV-infected testes, the concentrations of C1q and C3 protein increased by

5- and 3-fold, respectively (Fig. 4a, b), while those of MASP1, MBL, CFB or CFH demonstrated slight or no increase (Fig. 4c–f). Since the classical pathway is initiated by the immune complex, anti-ZIKV, and anti-sperm antibodies were also measured. Anti-ZIKV antibodies, rather than anti-sperm antibodies were detected at 14 dpi (Supplementary Fig. S7f, g), further suggesting the activation of the complement system through the classical pathway mediated by anti-ZIKV antibodies.

Since spermatogenic cells are susceptible to ZIKV infection, we then tested whether spermatogenic cells were damaged by complement activation. IFA was performed to observe the expression and distribution of complement components. In testis sections from ZIKV-infected A6 mice, C1q first appeared in the interstitium at 7 dpi, and then concentrated into the seminiferous tubules at 14 dpi (Fig. 4g), accompanied by the entry of inflammatory cells (Fig. 4h). Furthermore, as shown by adjacent sections, the MAC showed costaining with the spermatogenic cell marker DDX4 (Fig. 4i), and these MAC+ DDX4+ spermatogenic cells were also positive for ZIKV antigen (Fig. 4j). Intriguingly, MAC staining usually showed a stronger signal at the edge of seminiferous tubules (Fig. 4i), while ZIKV staining was distributed evenly in the tubules (Fig. 4j), suggesting that the spermatogenic cells

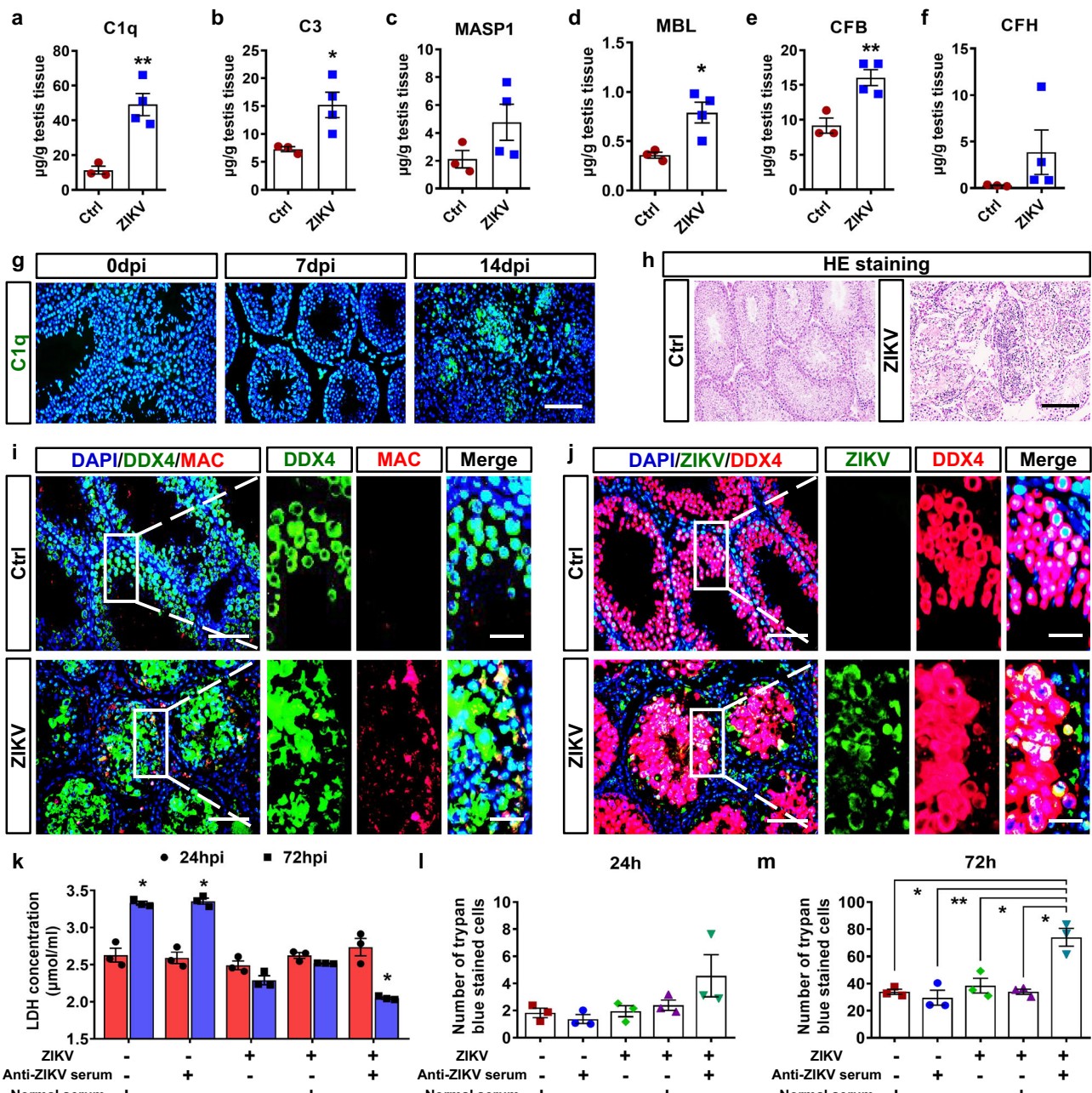

**Fig. 4 | The activation of complement system led to the damage of spermatogenic cells. a–f** Concentration of complement components in testes from control or ZIKV-infected A6 male mice at 14 dpi were analyzed by ELISA. Results were shown as means ± SEM ($n = 3$ mice for ctrl and 4 mice for ZIKV-infected) and analyzed using the two-sided Student's $t$ test. *$p < 0.05$, **$p < 0.01$. **g** The distribution of C1q was analyzed by IFA in ZIKV-infected testes from A6 mice at 0, 7, and 14 dpi. Scale bar, 25 μm. **h** HE staining of testes from control or ZIKV-infected A6 mice at 14 dpi. Scale bar, 50 μm. **i, j** The co-immunostaining of DDX4 and MAC (**i**), or DDX4 and ZIKV antigen (**j**) in testes from ZIKV-infected A6 mice. Nuclei were stained with

DAPI. Scale bar, 25 μm. (10 μm in enlarged panels). **k–m** The effects of complement system activation on ZIKV-infected JKT-1 cells were measured using an in vitro complement activation assay. Cells were incubated with guinea pig serum containing complement molecules for 30 min and stained with trypan blue for 2 min. Intracellular LDH concentration (**k**) and the number of trypan blue positive cells (see Supplementary Fig. S7h for representative pictures) was counted in each group at 24 and 72hpi (**l**, **m**, $n = 3$ independent experiment for each group). Results were shown as means ± SEM and analyzed using the two-sided Student's $t$ test. *$p < 0.05$. Exact $p$ values in Source Data file. Source data are provided as a Source Data file.

in basal compartments were more easily attacked by MAC, possibly because of their localization. MAC is the cytolytic effector of the complement system and can lead to osmolysis by forming pores in the plasma membrane of targeted cells[15]. The presence of MAC was therefore an index that spermatogenic cells were attacked by the activated complement system.

To further verify the role of classical complement activation in spermatogenic cell damage, we used the human seminoma cell line

JKT-1 to carry out an in vitro complement activation assay. In the experiment, ZIKV-infected or uninfected JKT-1 cells were incubated with anti-ZIKV antibodies in the presence of guinea pig serum which serves as a source of complement components[16]. At 24–72 h post infection, live cells were evaluated by measuring intracellular lactate dehydrogenase (LDH). The concentration of intracellular LDH significantly decreased in the presence of both ZIKV and anti-ZIKV serum at 72 h post infection (Fig. 4l). In parallel, cells attacked by complement

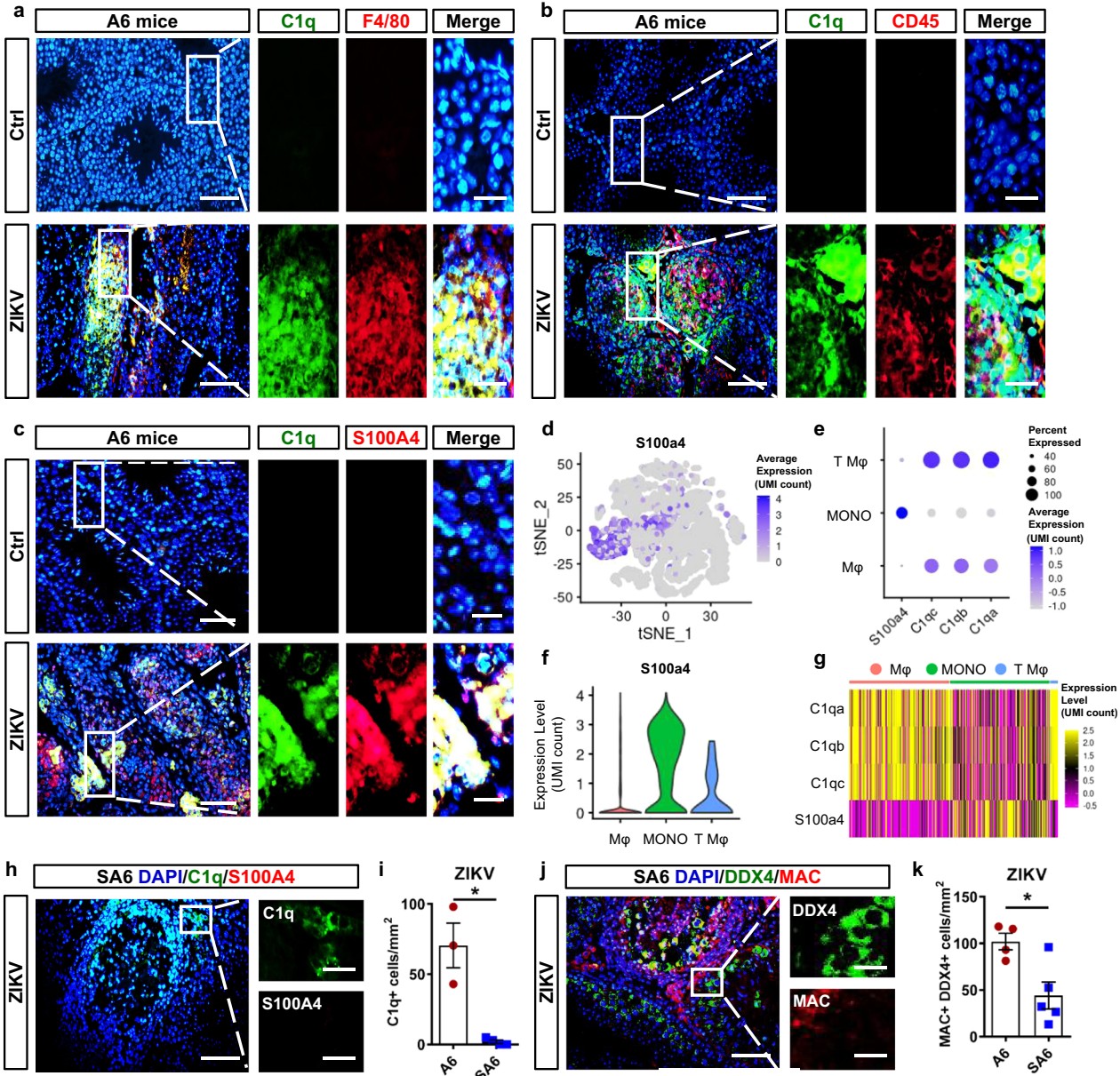

**Fig. 5 | C1q was primarily produced by S100A4+monocytes/macrophages.**
**a**–**c** The co-immunostaining of C1q and F4/80 (**a**), C1q and CD45 (**b**), or C1q and S100A4 (**c**) in testes from control or ZIKV-infected A6 mice at 14 dpi. Scale bar, 25 μm. (10 μm in enlarged panels). **d** The distribution of *S100a4*-expressing cells in tSNE chart. **e** The expression level of *S100a4*, *C1qa*, *C1qb*, *C1qc* in exogenous macrophages (Mφ), monocytes (MONO), and testicular macrophages (T Mφ) in bubble chart. **f**, **g** The expression level of *S100a4* in exogenous macrophages (Mφ), monocytes (MONO), and testicular macrophages (T Mφ) were shown in violin chart

(**f**) and heatmap (**g**). **h**–**k** The co-immunostaining of C1q and S100A4 (**h**), and DDX4 and MAC (**j**), in testes from control or ZIKV-infected SA6 mice at 14 dpi. C1q+ cells (**i**, *n* = 3 mice for ctrl and 4 mice for ZIKV-infected), or MAC+ DDX4+ cells (**k**, *n* = 4 mice for ctrl and 5 mice for ZIKV-infected), were quantified and expressed as cells/ mm². Results were shown as means ± SEM and analyzed using the two-sided Student's *t* test. *\*p* < 0.05. Scale bar, 25 μm. (10 μm in enlarged panels). Exact *p* values in Source Data file. Source data are provided as a Source Data file.

activation were also visualized using trypan blue staining and then quantified. At 72 h post infection, the trypan blue-positive cell number significantly increased in cell culture with both ZIKV and anti-ZIKV serum (Fig. 4l, m and Supplementary Fig. S7h). The classical complement activation mediated by ZIKV antigen-antibody complexes, instead of ZIKV infection itself, was, therefore, the main cause for the damage to ZIKV-infected spermatogenic cells.

**S100A4+macrophages were the main source of C1q**
ScRNA-Seq suggested that macrophages were the main source of *C1q*. These results were also checked in testis sections and cell culture. As revealed by IFA, C1q staining showed no signal in uninfected testicular

tissues (Fig. 5a) or in primary testicular cell culture in vitro regardless of ZIKV infection (Supplementary Fig. S8a, b), suggesting that C1q was not secreted by testicular resident cells. In contrast, C1q was costained with the macrophage marker F4/80 and myeloid cell marker CD45 (Fig. 5a, b). These results were consistent with scRNA-Seq results that C1q was mainly expressed by exogenous macrophages (Fig. 3f–h), and they both indicated myeloid macrophages as the source of C1q.

Our previous study showed that a S100A4+ subpopulation of myeloid macrophages played crucial roles in ZIKV infection in the testis[5]. The contribution of S100A4+ macrophages to C1q expression was then investigated. While IFA did demonstrate colocalizations between C1q and S100A4 in ZIKV-infected testes (Fig. 5c), scRNA-Seq

indicated that *S100a4* transcripts were much higher in monocytes than macrophages (Fig. 5d, e) and negatively correlated with those of *C1qa*, *C1qb*, and *C1qc* (Fig. 5f, g and Supplementary Fig. S8c). These results implied that the C1q-expressing macrophages were differentiated from S100A4-expressing monocytes and that the *S100a4* gene was downregulated during differentiation, but the duration of S100A4 protein rendered the C1q-expressing cells positive for S100A4 staining.

To validate the contribution of S100A4+ monocytes/macrophages to C1q production, ZIKV-induced pathological changes and complement activation in the testis were analyzed using S100A4 deficient SA6 mice (*S100a4^{-/-} Ifnar^{-/-}* mice). Compared with those of A6 mice, the testes of ZIKV-infected SA6 mice were larger in size, heavier in weight (Supplementary Fig. S8d, e), and had fewer inflammatory cells and more intact seminiferous tubules (Supplementary Fig. S8f). The number of C1q+ cells and MAC+ spermatogenic cells, as revealed by coimmunostaining, was significantly reduced in ZIKV-infected SA6 testes (Fig. 5h–k). Moreover, the distribution of MAC+ cells was restricted to the edge of seminiferous tubules (Fig. 5j), suggesting that S100A4+ monocytes/macrophages were not only the source of C1q but also able to promote the penetration of C1q into the lumen of seminiferous tubules. These results demonstrated an inhibition of complement activation in the testis by depleting S100A4 and indicating that S100A4+ monocyte-derived macrophages were the main source of C1q.

### Similar pathogenesis was observed in ZIKV-infected hSTAT2-KI mice

The A6 mice have a deficiency in IFN-I signaling so it's reasonable to question the representativeness of their pathogenesis. To clarify whether the above changes also occur in immunocompetent mice, we further analyzed the pathogenesis of ZIKV-infected testes using human STAT2 knock-in (hSTAT2 KI) mice, which were reported to be immunocompetent mice and susceptible to ZIKV infection[17].

Human STAT2 KI mice were intraperitoneally challenged with $1 \times 10^4$ ZIKV. Although the seminiferous tubules were relatively integrated into comparison to those of A6 mice, HE staining did show the interstitial infiltration of inflammatory cells and loosely arranged spermatogenic cells (Fig. 6a). IFA confirmed the loss of spermatogenic cells in some seminiferous tubules (Fig. 6b, c), and viral RNA was also detected in ZIKV-infected testes from hSTAT2 KI mice (Fig. 6d). ZIKV antigens were detected in both the testicular interstitium and intraluminal area and their intensities increased with infection progression (Fig. 6e), indicating that ZIKV did infect and cause damage in the testes of hSTAT2 KI mice.

We next investigated the activation of the complement system and its effects on spermatogenic cells in hSTAT2 KI mice. Upregulation of *C1q*, *C3*, and *S100a4* RNA was detected in ZIKV-infected testes, while *Cfb*, *Cfp*, *Masp1*, or *Mbl1* demonstrated slight or no increase (Fig. 6f–l). ELISA also verified the upregulated expression of C1q (Fig. 6m). By coimmunostaining, C1q showed colocalization with S100A4 and F4/80, but the intensity of C1q was weaker than that observed in A6 mice (Fig. 6n–p). Notably, MAC was costained with DDX4, the molecular marker of spermatogenic cells (Fig. 6q), and the number of DDX4+ MAC+ spermatogenic cells significantly increased after infection (Fig. 6r). These results indicated that, although to a lesser extent, the pathogenesis of ZIKV-infected testes from hSTAT2 KI mice showed similar characteristics to that of A6 mice (Fig. 4h), and thus, activation of the classical complement pathway is a common pathway in ZIKV-induced testicular injury.

### MAC attacked spermatogenic cells in nonhuman primates

Because A6 mice are deficient in the IFN-α/β signaling pathway and their pathogenesis after ZIKV infection might be different from that of humans, we performed IFA using testes from ZIKV-infected northern

pigtailed macaques (*macaca leonine*) to test whether the complement system was similarly activated in ZIKV-infected nonhuman primates. In ZIKV-infected macaques at 60 dpi, testicular pathological changes including expansion of the interstitial space and infiltration of inflammatory cells in seminiferous tubules were clearly observed (Fig. 7a). ZIKV antigens were also detected in a few spermatogenic cells and Sertoli cells (Fig. 7b). Moreover, MAC+ spermatogenic cells and C1q+ S100A4+ monocyte/macrophages were all present in testes from ZIKV-infected macaques, as revealed by coimmunostaining assay (Fig. 7c, d). Quantification further showed that DDX4+ spermatogenic cells significantly decreased after ZIKV infection whereas DDX4+ MAC + spermatogenic cells, C1q+ cells, and S100A4+ cells were all increased (Fig. 7e–h), displaying features similar to those in A6 mice. Taken together, although ZIKV-induced testicular pathological changes were substantially milder in macaques, they did demonstrate infiltration of S100A4+ monocyte/macrophages, activation of the complement system, and loss of spermatogenic cells, implying that activation of the complement system in the testis is a common response in ZIKV-infected mice and macaques.

The testes from ZIKV-infected macaques were subjected to RNA sequencing for further analysis. Among 1963 upregulated genes and 1288 downregulated genes (Fig. 7i), complement activation-related genes such as *C1s*, the executor of the catalytic function of the classical pathway[18], and *C1r*, the enzyme responsible for intrinsic activation of the C1 complex[19] were all upregulated. Macrophage regulation-related genes and myeloid macrophage-related genes were also upregulated, which was consistent with the infiltration of myeloid macrophages in ZIKV-infected testes (Fig. 7j). In contrast, testicular cell markers genes were decreased (Fig. 7j), indicating spermatogenic cell damage in ZIKV-infected macaques.

As revealed by Kyoto Encyclopedia of Genes and Genomes (KEGG) analysis and Gene Ontology (GO) enrichment analysis, although the genes in inflammatory pathways showed upregulation tendency (Supplementary Fig. S9a), most upregulated genes belonged to pathways associated with cellular proliferation, survival, and tissue repair, such as the cell periphery, cell communication, cell adhesion, and biological adhesion (Supplementary Fig. S9b), showing a strong self-repair response in ZIKV-infected macaques. Taken together, these results implied that ZIKV infection leads to a long-term impact on the testes of macaques, even up to 60 dpi, which were characterized by features similar to those of A6 mice, such as macrophage infiltration, complement activation, and spermatogenic cell damage. However, possibly due to the intact IFN-α/β signaling pathway, macaques had stronger responses regarding cell junctions, cell survival, and tissue repair, which effectively restricted the damage to spermatogenic cells caused by complement activation, and made the pathological change in immune-competent macaques much milder than that in A6 mice.

### S100A4 inhibitors alleviated ZIKV-induced spermatogenic cell damage

The crucial roles of complement activation in ZIKV-induced damage to spermatogenic cells led us to test whether they were potential therapeutic targets for ZIKV infection. We first treated ZIKV-infected A6 male mice with a C1 inhibitor (C1INH). To mimic clinical therapy, ZIKV-infected A6 male mice were not administered C1INH until 5 dpi when they had visible manifestations. Compared to the mock-treated mice, testes from C1INH-treated mice were larger and heavier (Supplementary Fig. S10a, b). Their seminiferous tubules were more intact (Supplementary Fig. S10c), and the number of MAC+ spermatogenic cells was significantly reduced (Supplementary Fig. S10d, e), confirming that complement activation was an important reason for the damage in testes. Although C1INH alleviated the pathogenesis in ZIKV-infected testes, the viral loads in testes and blood were higher than those from mock-treated mice (Supplementary Fig. S10f, g), and the body weight loss and symptom score were more severe in C1INH-treated mice

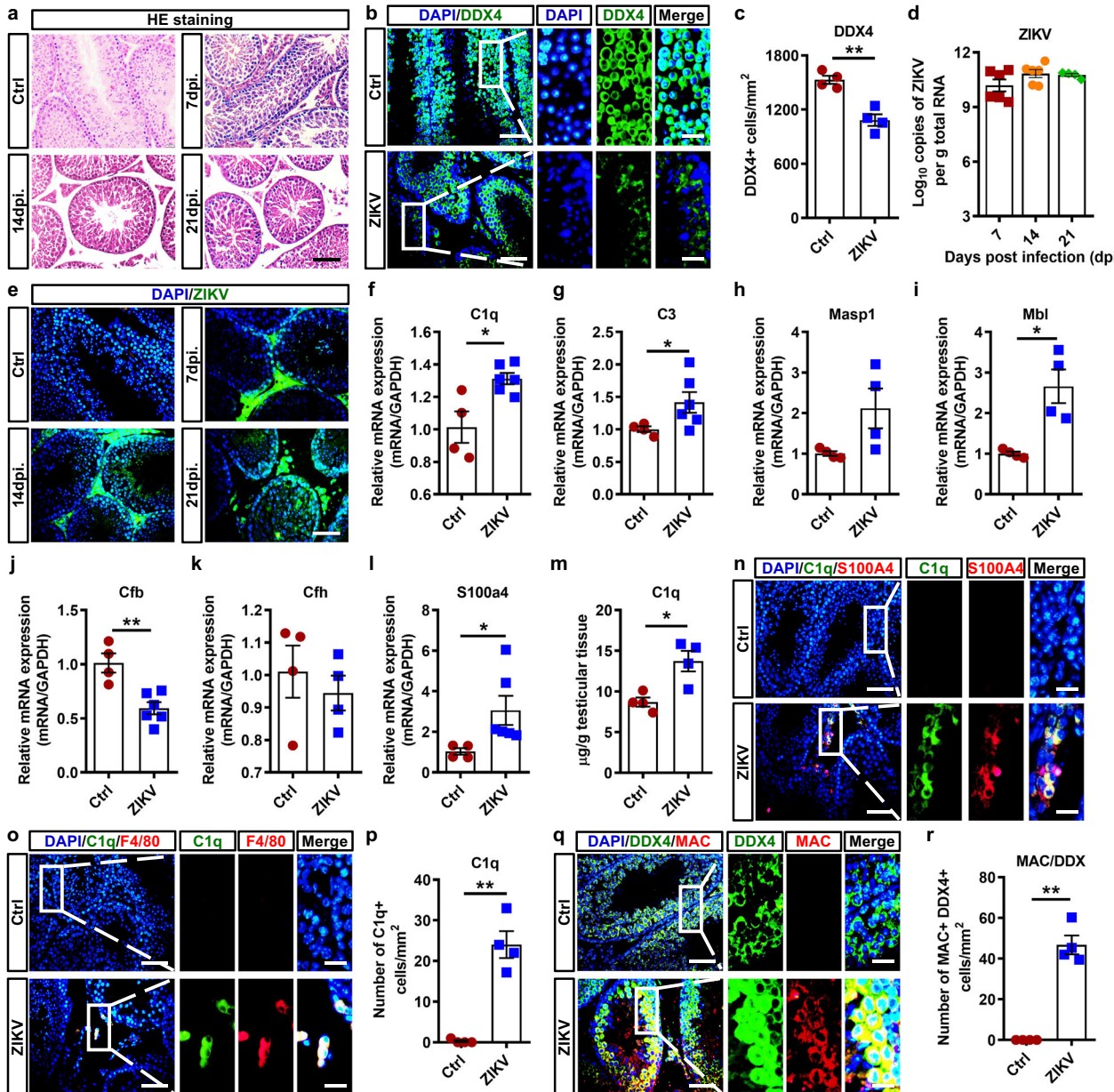

**Fig. 6 | ZIKV-induced spermatogenic cell damage and complement activation in testes from hSTAT2 KI mice. a** HE staining of testes from ZIKV-infected hSTAT2 KI mice. Scale bar, 25 μm. **b**, **c** IFA with anti-DDX4 antibody in testis from control or ZIKV-infected hSTAT2 KI mice at 14 dpi. **b** Number of DDX4+ spermatogenic cells/mm² (**c**) were shown as means ± SEM and analyzed using the two-sided Student's *t* test. **p < 0.01. (*n* = 4 testes for each group). Scale bar, 25 μm. (10 μm in enlarged panels). **d** ZIKV viral load in testes of ZIKV-infected hSTAT2 KI mice (*n* = 4 testes for each group). Results were shown as means ± SEM and analyzed using the two-sided Student's *t* test. **e** The distribution of ZIKV antigens in testis from control or ZIKV-infected hSTAT2 KI mice. Scale bar, 25 μm. **f–m** The relative RNA expression of complement components (**f–k**) and *S100a4* (**l**) and the concentration of C1q protein (**m**) in testes from control or ZIKV-infected hSTAT2 KI male mice at 14 dpi.

Results were shown as means ± SEM and analyzed using the two-sided Student's *t* test. *p < 0.05, **p < 0.01. (*n* = 4 testes for ctrl and 4–6 testes for ZIKV-infected). **n**, **o** The co-immunostaining of C1q and S100A4 (**n**) or F4/80 (**o**) in testes from control or ZIKV-infected hSTAT2 KI mice at 14 dpi. Scale bar, 25 μm. (10 μm in enlarged panels). **p** Number of C1q+ cells/mm² was shown as means ± SEM and analyzed using the two-sided Student's *t* test. **p < 0.01. (*n* = 4 testes for each group). **q**, **r** The co-immunostaining of DDX4 and MAC (**q**) in testes from control or ZIKV-infected hSTAT2 KI mice at 14 dpi. MAC+ DDX4+ cells (**r**) were quantified and expressed as cells/mm². Results were shown as means ± SEM and analyzed using the two-sided Student's *t* test. **p < 0.01. (*n* = 4 testes for each group). Nuclei were stained with DAPI. Scale bar, 25 μm (10 μm in enlarged panels). Exact *p* values in Source Data file. Source data are provided as a Source Data file.

(Supplementary Fig. S10h, i), suggesting that blocking the activation of the classical pathway would prevent virus clearance and generally harm ZIKV-infected mice.

As demonstrated above, S100A4+ monocytes/macrophages were the main source of C1q in the testis, and we then tested whether S100A4 inhibitors could be used to alleviate ZIKV-induced spermatogenic cell damage. Niclosamide and sulindac are two FDA-approved medicines and have been reported as S100A4 inhibitors[20,21]. Similarly,

niclosamide or sulindac was not administered to ZIKV-infected A6 male mice until 5 dpi. Their inhibitory effect on S100A4 expression was validated in bone morrow, spleen, and testis sections by IFA. Niclosamide rather than sulindac effectively inhibited the expression of S100A4 (Supplementary Fig. S11a), so niclosamide was chosen for further analysis. As expected, the inhibition of S100A4 decreased the expression of *C1q* and *C3* in ZIKV-infected testes (Fig. 8a, b). In line with this, the testes from niclosamide-treated mice had a lower viral load

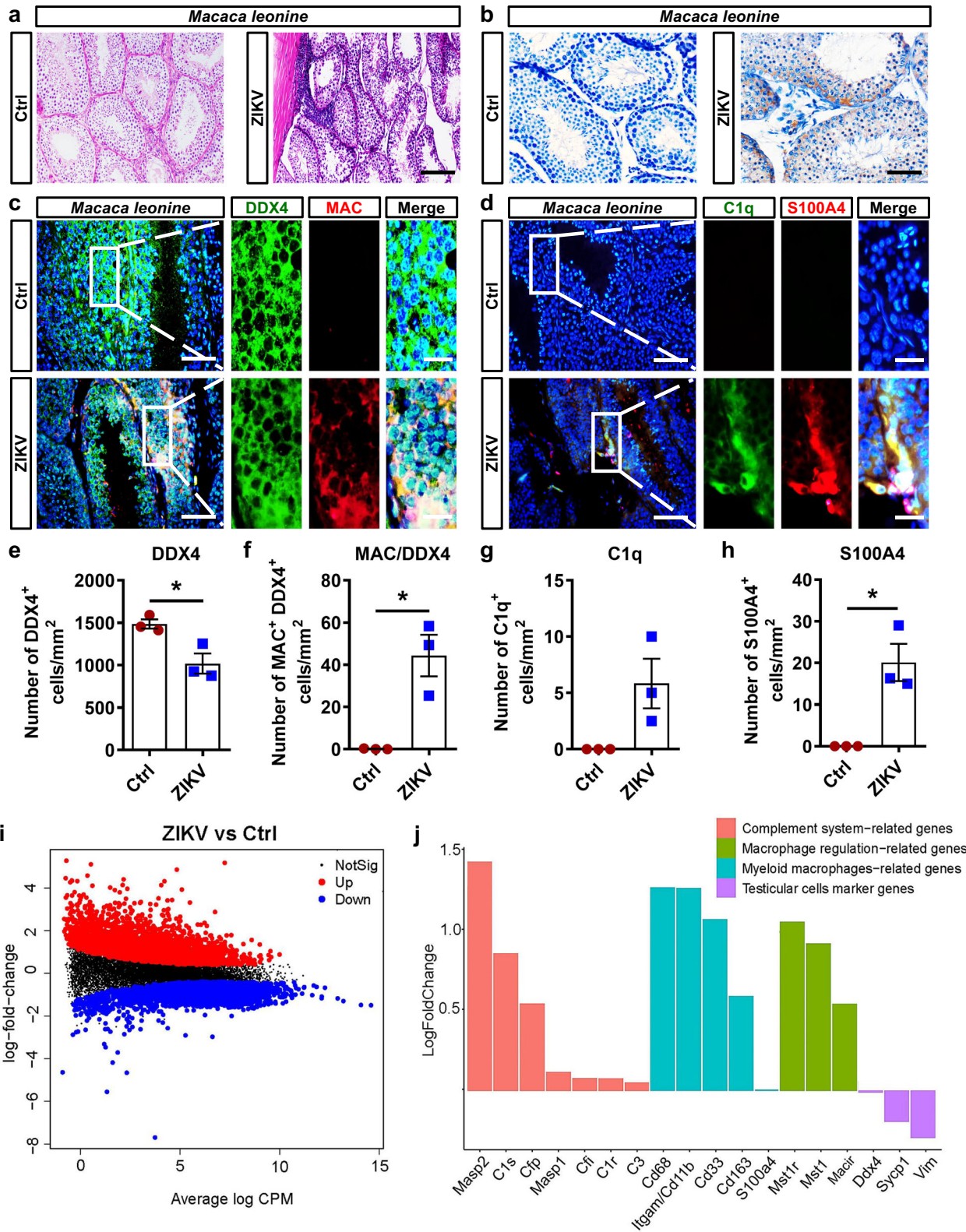

(Supplementary Fig. S11b), heavier weight, fewer inflammatory cells, less severe seminiferous tubule morphology, fewer MAC+ cells and more spermatogenic cells (Fig. 8c–g and Supplementary Fig. S11c), indicating the protective effects of niclosamide on ZIKV-infected testes. Meanwhile, unlike the side effects of C1INH, niclosamide-treated mice had a tendency of viral load reduction in many organs, and similar body weight changes and symptom scores

(Supplementary Fig. S11d–i), suggesting that niclosamide could alleviate ZIKV-induced testicular damage by inhibiting S100A4 expression and subsequent complement activation.

The fertility of niclosamide-treated A6 male mice was analyzed by mating them with healthy females. Uninfected mice and mock-treated mice were used as controls. After ZIKV infection, mock-treated A6 male mice showed a low fertility rate of 20% (Fig. 8h). In contrast, the

**Fig. 7 | Complement activation resulted in the damage of spermatogenic cells in testes from ZIKV-infected *macaca leonine*. a** HE staining of testes from ZIKV-infected macaca leonina at 60 dpi. Scale bar, 50 µm. **b** Distribution of ZIKV antigens in testes from ZIKV-infected macaca leonina at 60 dpi was shown by IHC staining. Scale bar, 25 µm. **c–h** The co-immunostaining of DDX4 and MAC (**c**), or C1q and S100A4 (**d**) in testes from control or ZIKV-infected *macaca leonine* at 60 dpi. DDX4+ cells (**e**), or MAC+ DDX4+ cells (**f**), or C1q+ cells (**g**), or S100A4+ cells **h** were quantified and expressed as cells/mm². Results were shown as means ± SEM and analyzed using the two-sided Student's *t* test. *$p < 0.05$. ($n = 3$ *macaca leonine* for each group). Nuclei were stained with DAPI. Scale bar, 25 µm (10 µm in enlarged panels). **i, j** Testes from control or ZIKV-infected *macaca leonine* were isolated at 60 dpi and subjected to RNA-sequencing. Plot of log fold change vs. average log CPM (log counts per million) of all detected transcripts. Points are colored according to expression status: non-significant genes (black), significant upregulated genes, red; and downregulated genes, blue (**i**). Log fold change of 18 differentially expressed genes (**j**). ($n = 3–4$ *macaca leonine* for each group). Exact *p* values in Source Data file. Source data are provided as a Source Data file.

fertility rate of niclosamide-treated mice increased to 60% (Fig. 8h), and their offspring, although fewer than the uninfected mice (Fig. 8i), developed normally like those born to the uninfected mice (Fig. 8j). Taken together, these results demonstrated that niclosamide could save the fertility of male mice from ZIKV infection by inhibiting S100A4 expression and subsequent complement activation.

## Discussion

ZIKV, a flavivirus usually transmitted by mosquito bites[22,23], can also establish long-term infection in the testis and consequently transmit sexually from male to female[24]. Moreover, relevant symptoms such as oligospermia and hematospermia have been reported in many male ZIKV patients, making ZIKV a potential threat to male reproductive health. To analyze the mechanism underlying ZIKV-induced spermatogenic cell damage, we performed scRNA-Seq on testicular tissues from healthy and ZIKV-infected mice. The scRNA-Seq allows researchers to identify cell clusters and characterize them separately[25], which gave us an opportunity for deep insight into ZIKV infection in each type of testicular cell. By using scRNA-Seq, we measured the presence of ZIKV RNA in various testicular or inflammatory cells and evaluated the impact of ZIKV infection on the gene expression of these cells.

The testis is an important target organ for ZIKV[26,27]. Although many advances have been achieved in the field[1,28,29], the detailed mechanism has not yet been completely addressed. Using scRNA-Seq, we found that spermatogonia, which are the stem cells of spermatogenesis, were highly susceptible to ZIKV infection. This result shed new light on the mechanisms underlying ZIKV infection in the testis. Structurally, spermatogonia are located in the basal compartment and remain outside the BTB[30], which renders them more accessible to the virus in circulation. When spermatogonia differentiate into spermatocytes, they can regulate the opening of the BTB to allow spermatocytes to enter the seminiferous tubules to continue further differentiation[31,32]. In this way, spermatogonia themselves can act as a Trojan horse to bring the virus into the seminiferous tubules.

In a previous study, we found that myeloid-derived S100A4 + macrophage infiltration is a characteristic of ZIKV infection in the testes of mice with IFNRA knockout[5]. In this research, we further showed that myeloid-derived S100A4 + macrophages, strictly speaking, should be S100A4 + monocytes. When they were recruited into ZIKV-infected testes, the expression of S100A4 was downregulated, while the expression of complement factors was upregulated. The complement components secreted by these cells, as well as other cells, were activated through the classical pathway mediated by anti-ZIKV antibodies and led to the damage of spermatogenic cells by forming MAC on the cell surface. Therefore, monocyte/macrophage infiltration, complement activation, and spermatogenic cell damage are the main pathological characteristics of ZIKV-infected testes.

The complement system is a well-known component of the immune system and plays important roles in the clearance of viruses such as Chandipura virus (CHPV), H1N1 influenza virus, and influenza B virus[33,34]. Its overactivation is also reported to enhance the severity of some infectious diseases, including sepsis, hepatitis C virus infection, and alphavirus-induced rheumatic disease[35–37],

usually via the release of large amounts of inflammatory cytokines. ZIKV-induced anti-C1q antibody responses have been reported to contribute to neurological complications in ZIKV-infected mice and monkeys[38]. In the testis, spermatogenic cells have been proven to lack C3 convertase regulators and thus are prone to attack by the complement system[39]. Subsequent studies have demonstrated that the abnormal activation of the complement system could lead to severe histological degeneration in the testis[40]. Here, we demonstrated another pathway by which activation of the complement system promoted pathogenesis. Spermatogenic cells normally develop in an immune-privileged environment and are hardly attacked by the immune system. However, when the BTB is disrupted by IFN-γ secreted by S100A4 + monocytes/macrophages[5], spermatogenic cells are exposed to the complement system and, due to the lack of complement inhibitory factors, more easily attacked by the activated complement system.

Regarding the pathological change in ZIKV-infected testes, the representativeness of mice with IFNAR knockout is a reasonable concern. As clinical evidence shows, although oligospermia and hematospermia have been well documented and infectious particles or viral RNA have been detected in the semen of ZIKV patients for a long time[41,42], severe symptoms such as orchitis are rarely reported[43]. In this research, we used hSTAT2 KI mice and northern pigtailed macaques to address this concern because they can better mimic the pathogenesis of ZIKV in humans[17,44,45]. As revealed by IFA and RNA sequencing, although the testicular pathological change was mild in ZIKV-infected hSTAT2 KI mice and macaques, they did show characteristics similar to those observed in A6 mice, including the infiltration of S100A4 + monocytes/macrophages, complement activation, and spermatogenic cell damage. These results suggest that the lesions observed in A6 mice are similar to those in immunocompetent models. Nevertheless, in the presence of an intact IFN-α/β signaling pathway, hSTAT2 KI mice and macaques had much stronger responses regarding cell junctions, cell survival, and tissue repair, which effectively restricted the damage to spermatogenic cells.

Based on the role of classical complement activation in ZIKV-induced testis damage, we tested its inhibitor C1INH for the treatment of ZIKV infection. Although C1INH effectively alleviates testicular damage by inhibiting MAC synthesis by blocking classical complement activation, the viral load of the testis significantly increased after C1INH treatment. The complement system has been proven to reduce Zika viral load via the formation of the MAC[46], and it is likely that inhibiting the complement system may have the risk of aggravating ZIKV infection. Therefore, the application of complement inhibitors in the treatment of ZIKV-infected men is not a good choice.

Since S100A4 monocytes/macrophages are the main producers of complement components, we further tested the feasibility of using S100A4 + monocytes as drug targets and found that the S100A4 inhibitor niclosamide could effectively alleviate ZIKV-induced testicular damage and protect the fertility of male mice after ZIKV infection. Regarding sulindac, although it was reported to interdict S100A4 synthesis by intervening in β-catenin signaling[20], it cannot block other possible synthesis pathways of S100A4 protein, and this might be the reason why its blocking effect is not as good as niclosamide.

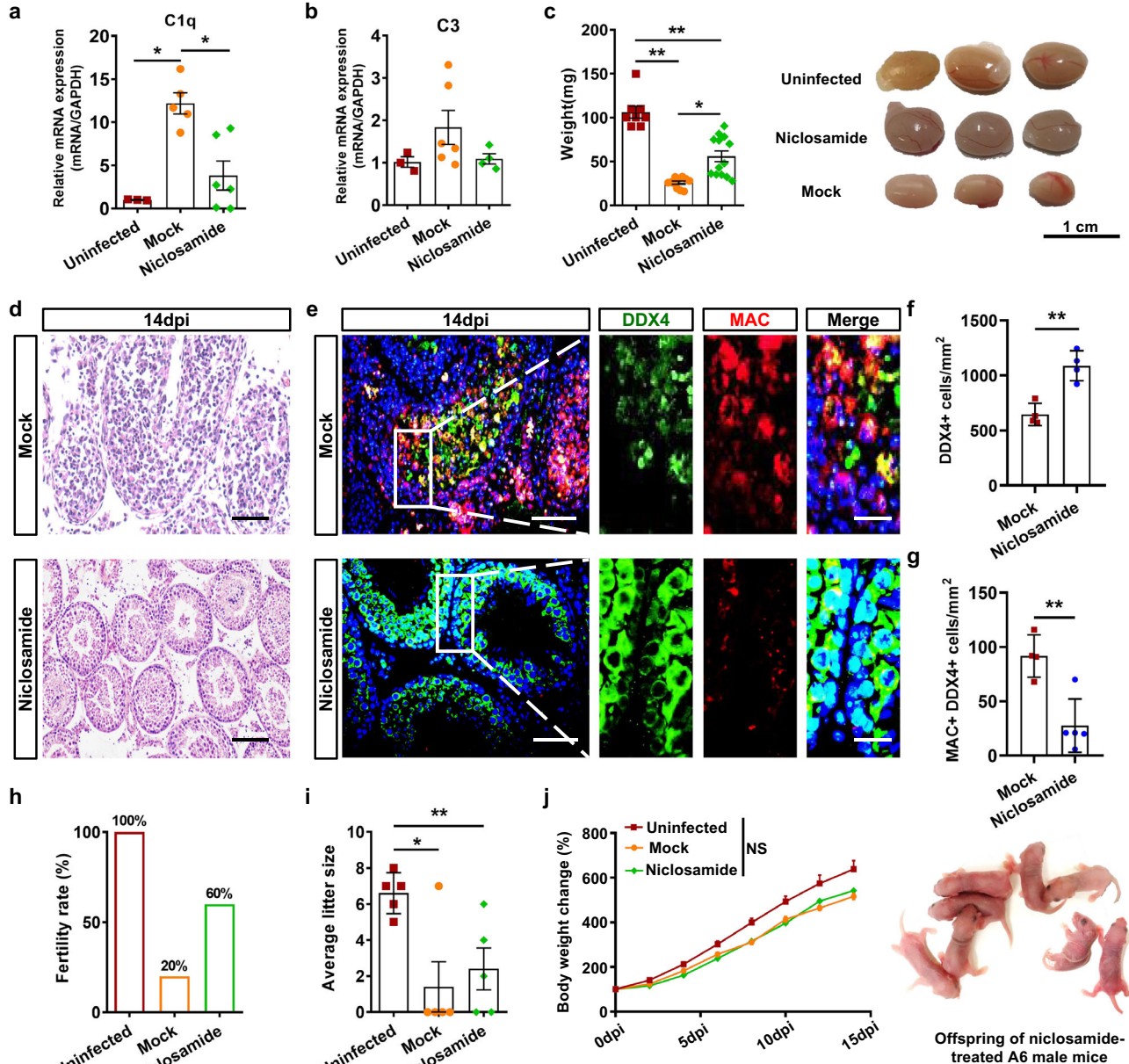

**Fig. 8 | S100A4 inhibitor niclosamide alleviated ZIKV-induced spermatogenic cells damage. a, b** The relative mRNA expression levels of *C1q* (**a**), and *C3* (**b**) in testes at 14 dpi were measured by RT–qPCR. Results were shown as means ± SEM and analyzed using the two-sided Student's *t* test. *$p < 0.05$. ($n = 3$ mice for uninfected, 5–6 mice for mock-infected, and 4–6 mice for niclosamide-treated). **c** The weight and representative pictures of testes in at 14 dpi. Results were shown as means ± SEM and analyzed using the two-sided Student's *t* test. *$p < 0.05$, **$p < 0.01$ ($n = 8$ testes for uninfected, $n = 11$ testes for mock-infected, and 13 testes for niclosamide-treated). **d** HE staining of testicular sections at 14 dpi, Scale bar, 50 μm. **e–g** Co-immunostaining of DDX4 and MAC in testicular sections at 14 dpi (**e**). Nuclei were visualized with DAPI. Scale bar, 25 μm (10 μm in enlarged panels). DDX4+ cells (**f**, $n = 4$ mice for each group) or MAC+ DDX4+ cells (**g**, $n = 4$ mice for mock-infected

and 5 mice for niclosamide-treated) were quantified and expressed as cells/mm². Results were shown as means ± SEM and analyzed using the two-sided Student's *t* test. **$p < 0.01$. **h–j** Fertility rate of male mice in each group (**h**). The average litter size of health females mated with male mice in each group (**i**, $n = 5$ litters for each group). Results were shown as means ± SEM and analyzed using the two-sided Student's *t* test. *$p < 0.05$, **$p < 0.01$. Neonatal born to females mated with niclosamide-treated A6 male mice and the body weights of offspring of male mice in each group (**j**, $n = 4$ sucking mice for uninfected, 7 sucking mice for mock-infected and six sucking mice for niclosamide-treated). Results were shown as means ± SEM and the comparison of body weight between the two groups were analyzed using repeated-measures ANOVA. *$p < 0.05$, **$p < 0.01$. Exact *p* values in Source Data file. Source data are provided as a Source Data file.

Niclosamide is an FDA-approved anti-helminthic drug and its safety has been well-tested. In an intrasplenically xenografted mouse model, niclosamide was shown to inhibit the expression of S100A4 and thus used as an S100A4 inhibitor for tumor therapy[21]. The inhibitory effect of niclosamide on S100A4 expression may be caused by blocking the S100A4 gene promoter, as no decrease in S100A4 expression was observed in the CRC cell line with S100A4 overexpression driven by the CMV promoter after niclosamide treatment[21]. Intriguingly,

niclosamide has also been found to reduce the infections of various viruses such as SARS-CoV, MERS-CoV, ZIKV, and human adenovirus[47]. In our study, however, niclosamide showed no or little effect on ZIKV infection in vivo. Its protective role is mainly attributed to the inhibition of S100A4 expression. On the basis of niclosamide, we believe that the development of a derivative carrying both antiviral and anti-S100A4 potential will provide better protection for male health during ZIKV infection.

## Methods

### Research compliances

All animal experiments in this study were conducted reviewed and approved by the Institutional Animal Care and the Animal Ethics Committees of the Capital Medical University (Approval number: AEEI-2015-049).

### Virus strain

ZIKV (strain SMGC-1, GenBank accession number: KX266255) was isolated in Shenzhen, China from an imported ZIKV patient in 2016. ZIKV was propagated in C6/36 cells. The titer was determined by plaque assay on Vero cell monolayer under overlay medium containing 1.0% methylcellulose.

### Cells

Vero cells (African green monkey kidney cells, National Infrastructure of Cell Line Resource, Cat#3111C0001CCC000060) were cultured in minimum essential medium (MEM, Gibco, USA) with 5% fetal bovine serum (FBS, PAN, Germany) at 37 °C. C6/36 cells (Aedes albopictus cells, National Infrastructure of Cell Line Resource, Cat#3131C0001000400021) were cultured in RPMI 1640 (Gibco, USA) with 10% FBS at 28 °C. JKT-1 cells (human seminoma cell line, HTX2280) were purchased from MLBIO, China, and cultured in Dulbecco's Modified Eagle Medium (DMEM, Gibco, USA) with 10% FBS at 37 °C.

### Animals

Balb/c mice were purchased from Vital River Laboratories (Beijing, China). hSTAT2 KI mice were kindly provided by the Jackson Laboratory. A6 mice were provided by the Institute of Laboratory Animals Science, Peking Union Medical College. SA6 mice were generated and characterized as previously reported[5]. All these mice were on C57BL/6 genetic background and bred under specific pathogen-free conditions. All mice were fed standard chow (irradiated cobalt-60) and housed in groups of up to five. Lighting was time controlled on a standard condition (12/12 h light/dark cycle). Temperature and humidity were stable and consistent at 20−26 °C and 40−70%, respectively. *Macaca leonine* were fed in Kunming Institute of Zoology, Chinese Academy of Sciences, Kunming, China.

### Animal experiments

6−8-week-old male mice, including A6, SA6, and hSTAT2 KI mice, were intraperitoneally (i.p.) challenged with $10^4$ pfu ZIKV or PBS. To investigate the effect of C1INH or S100A4 inhibitors on ZIKV-induced testicular damage, ZIKV-infected A6 male mice were i.p. injected with 0.1 mg per mouse of C1INH or 20 mg/kg body weight of niclosamide or sulindac or solvent alone (Mock) from 5 dpi till euthanasia or death, respectively. Blood, sera, and organs from ZIKV-infected male mice were isolated at 7, 10, 14, 21, and 28 dpi. Body weight, disease symptoms, and survival rates of mice were recorded daily till euthanasia or death. Five-year-old adult male *macaca leonine* was subcutaneously challenged with $10^6$ pfu ZIKV or PBS. Testes were isolated at 60 dpi from the *macaca leonine*. In order to investigate the fertility of male mice in each group, ZIKV-infected male mice at the recovery stage were co-caged with uninfected female mice with a ratio of 1:1. The pregnancy and fertility of female mice were observed within 30 days after being co-caged.

### ZIKV mRNA quantification and complement system-related molecules mRNA relative quantification

S100A4 inhibitors-treated, ZIKV-infected and control mice were euthanized and whole blood and major organs were harvested as indicated. Samples were homogenated in Trizol (Transgen China, ET101-01) and RNA was extracted according to manufacturer protocol. Real-time qPCR analyses were performed as previously reported[48] with Quant One Step qRT-PCR (Tiangen, China) on 7500 Real-Time PCR

System (Applied Biosystems, USA). Quantification of the copies of ZIKV mRNA was determined using the standard curve method. The ZIKV genome RNA transcribed in vitro was quantified and used as a standard template to establish the standard curve[48]. Relative quantification of complement system-related molecules mRNA was determined by using GAPDH as internal control and $2^{-\Delta\Delta Ct}$ method. The primer sequences were as follows: ZIKV forward: 5′-TCAGACTGCGA CAGTTCGAGT-3′; ZIKV reverse: 5′-GCATATTGACAATCCGGAAT-3′; GAPDH forward: 5′-GCATTGTGGAAGGGCTCA-3′; GAPDH reverse: 5′-ACCAGTGGATG CAGGGAT-3′; mS100A4 forward: 5′-AGCTGCATTC CAGAAGGTGA-3′, mS100A4 reverse 5′-CCCACTGGCAAACTACACCC-3′[5]; C1q forward: 5′-CGGGTCTCAAAGGAGAGAGA-3′; C1q reverse: 5′-TATTG CCTGGATTGCCTTTC-3′[49]; MBL1 forward: 5′-CTTCTGCTT CCATTACT CCCTG-3′; MBL1 reverse: 5′-GCCTTTTGGTCCTGGA CTTCC-3′; MASP1 forward: 5′-AC TTTCCGGTCAGA TTTCTCCA-3′; MASP1 reverse: 5′-TAGCCACCGATGTAGTTGTGA-3′; Complement factor B (CFB) forward: 5′-GAAAGCCAGTTGTGAGAGAG-3′; CFB reverse:5′-AGAGGAACCGTGGGAGTGA-3′[50]; Complement factor P (CFP) forward: 5′-GGTAAAGTAGCCAACGAG TC-3′; CFP reverse: 5′-GCAGA AGGGTTCAGTAGTAG-3′[50].

### ELISA

The concentrations of complement system-related molecules in testes from ZIKV-infected A6 mice were quantified by ELISA according to the manufacturer's instructions. C1q, Masp1, MBL, CFB, complement factor H (CFH), and C3 reagents were purchased from Cloud-Clone (mouse C1q, Masp1, MBL, CFB, CFH and C3 ELISA Kits, Cloud-Clone, China). Anti-sperm antibodies (Asab) reagents were purchased from MLBIO, China. Briefly, 100 μl standard or testis tissue supernatants (100 μl/well) was added into plates and incubated for 1 h at 37 °C. Subsequently, 50 μl Detection Reagent A was added and incubated for 1 h at 37 °C. After washing, 50 μl Detection Reagent B was added and incubated for 30 min. Finally, 90 μl Substrate Solution was added and incubated at 37 °C for 15 min, followed by the addition of 50 μl Stop Solution. The absorbance was measured at 450 nm in a Multiskan spectrum 1500 (Thermo, USA).

The anti-ZIKV antibodies in mouse serum were detected according to our previous study[51]. Briefly, the solution containing ZIKV amplified in C6/36 cells was centrifuged at 4 °C for 30 min at $3000 \times g$, and the supernatant was transferred to a 50 ml centrifuge tube, mixed with 40% PEG-8000-NTE buffer and incubated at 4 °C overnight. The next day, the mixture was centrifuged at $12,000 \times g$ and 4 °C for 30 min to recover the pallet containing viral protein. Resuspended the viral protein with borate buffer, quantify the protein concentration by nanodrop 2000, and stored at −80 °C. ZIKV protein was diluted to 100 μg/ml with carbonate coating buffer, then added into 96-well plate with 100 μl/well, and incubated overnight at 4 °C. After washed with PBST, plate was blocked by 1% BSA-PBS and incubated at 37 °C for 2 h. For ELISA assay, the serum from ZIKV-infected or uninfected A6 male mice at 7, 14, 21, and 28 dpi were diluted by PBST with the dilution of 1:10−1:12800. Mouse serum contain anti-ZIKV antibodies was used as the positive control, and 0.1% BSA-PBS was used as the blank control. Plate was incubated at 4 °C overnight, washed with PBST, and incubated with goat anti-mouse HRP-labeled secondary antibodies (1/100,000, Abcam, ab6789) at 37 °C for 1 h. The reaction was visualized by adding the chromogenic reagent (2.57 ml 0.2 M $Na_2HPO_4$, 2.43 ml 0.1 M citric acid, 5 ml $ddH_2O$, OPD (o-phenylenediamine) 5 mg, 15 mg μL hydrogen peroxide, mixed in dark;) and stopped by adding 2 M $H_2SO_4$. The absorbance was measured at 450 nm in a Multiskan spectrum 1500 (Thermo, USA).

### Virus infection in vitro

JKT-1 cells or primary testicular cells from A6 male mice were seed into 48-well plate with a cell density of $5 \times 10^5$ cells/ml and cultured at 37 °C for 18 h. After removing non-adherent cells by washing with PBS, the

attached cells were infected with ZIKV at MOI = 1 at 37 °C. The cells were harvested 24–72 h after infection and subjected to immuno-fluorescence staining.

## Immunofluorescence staining (IFA)

The testis was fixed in Modified Davidson's Fluid solution (30 ml of 40% formaldehyde, 15 ml of ethanol, 5 ml of glacial acetic acid, and 50 ml of distilled water) overnight, while other tissues were fixed in 4% paraformaldehyde (PFA) solution overnight, then dehydrated and paraffin-embedded. JKT-1 cells or primary testicular cells from A6 male mice seed in 48-well plate were fixed in 4% PFA for 15 min. Sections (5 μm in thickness) or cells in plate were incubated with the following primary antibodies overnight at 4 °C, including rat anti-mouse C1q (1:100, Abcam, ab11861), rat anti-mouse C3 (1:100, Abcam, ab11862), rabbit anti-mouse MAC (1:100, Abcam, ab55811), rabbit anti-mouse CD45 (1:100, Abcam, ab10558), rabbit anti-mouse Granzyme B (GZMB) (1:100, Abcam, ab4059), rabbit anti-mouse F4/80 (1:100, Abcam, ab111101), rabbit anti-mouse Caspase-9 (1:500, Abcam, ab202068), rabbit anti-mouse Caspase-8 (1:500, Abcam, ab227430), rabbit anti-mouse Cleaved Caspase-3 (1:500, Cell Signaling Technology, 9664 S), rabbit anti-mouse Caspase-1 antibody (1:500, Abcam, ab74279), mouse anti-mouse DDX4 (1:100, Abcam, ab27591), rabbit anti-mouse DDX4 antibody (1:100, Abcam, ab13840), rabbit anti-mouse S100A4 antibody (1:500, Cell Signaling Technology, 13018 S), or mouse anti-ZIKV antibody 4G2 (1:500). After cleaned with PBS, sections were incubated with the following secondary antibodies at 37 °C for 1 h, including donkey anti-mouse IgG (1:1000, Alexa Fluor R 488, A21202, Life technologies), donkey anti-rabbit IgG (1:1000, Alexa Fluor R 594, A21207, Life technologies), goat anti-rabbit IgG (1:1000, Alexa Fluor 488, A-11008, Life technologies), goat anti-mouse IgG 594 (1:1000, Alexa Fluor, A-11005, Life technologies) or goat anti-rat IgG (1:1000, Alexa Fluor 488, ab150157). Images were captured with Olympus microscope (IX71, Olympus, Japan).

## Hematoxylin and eosin (HE) analyses

To investigate the pathological changes of ZIKV-infected testes, testicular sections were subjected to HE staining by immersing them into xylene and alcohol, then stained with hematoxylin for 12 min. After being stained with eosin for 20 min and re-immersed in alcohol and xylene, sections were mounted using synthetic resin.

## In vitro complement activation assay

JKT-1 cells were seed into 48-well plate with a cell density of $5 \times 10^5$ cells/ml, cultured at 37 °C for 18 h, and then infected with ZIKV at MOI = 1. At 24- and 72-hour post infection (hpi), ZIKV-infected or uninfected cells were incubated with 200 μl/well sera from ZIKV-immunized or non-immunized mice. After incubated at 37 °C for 30 min, cells were washed with PBS and then incubated with 200 μl/well guinea pig serum containing complement molecules for 30 min. After washed with PBS, cells were stained with 100 μl trypan blue for 2 min. Images were captured with Olympus microscope (IX71, Olympus, Japan).

Intracellular LDH concentration of residual cells in plates was quantified by lactate dehydrogenase (LDH) activity detection kit (Solarbio, BC0685, China) according to the manufacturer's instructions. Briefly, cells were collected into a 1.5 ml a centrifuge tube, then extraction solution (75 μl/well) was added. Cells were then broken by ultrasonic wave. After centrifuged at 4 °C for 10 min, 10 μl supernatant was taken and added into 50 μl Detection Reagent I and 10 μl Detection Reagent II. After incubated for 15 min at 37 °C, 50 μl Detection Reagent III was added and incubated for 15 min. Finally, 150 μl Detection Reagent IV was added and incubated at room temperature for 3 min, 200 μl/sample resulting mixture was added into 96-well plate. The absorbance was measured at 450 nm in a Multiskan spectrum 1500 (Thermo, USA).

## Cell quantification

Positive staining cells in IFA assay or trypan blue positive cells mentioned above were analyzed with Image J v1.8.0.112 software as previously reported[5]. Each group contained at least three mice. A section from each mouse was analyzed in at least five fields (×200) and each field contained more than 1000 cells. Cell number was expressed as cells/mm$^2$.

## Library preparation and RNA sequencing

Testes from ZIKV-infected *macaca leonine* were harvested at 60 dpi. Uninfected *macaca leonine* served as controls. Library construction and sequencing was carried out by Novogene (Beijing, China). Briefly, no less than 1 mg of RNA per sample was used as the input material for RNA sample preparation. Sequencing libraries were generated using NEBNext® UltraTM RNA Library Prep Kit for Illumina® (NEB, Ipswich, MA, USA) following the manufacturer's recommendations. After the library was constructed, Qubit2.0 Fluorometer was used for preliminary quantification. Then, the samples were diluted to 1.5 ng/ml, and an Agilent 2100 bioanalyzer (Agilent 2100 bioanalyzer, Agilent Technologies, Wokingham, Berks, UK) was used to detect the insert size of the library.

The library preparations were sequenced using the Illumina sequencing platform (NovaSeq 6000, Illumina, San Diego, California, USA). For quality control, we used Fastp v0.23.2[52] to obtain clean data by removing reads containing adapter, reads containing ploy-N (>10%) and reads containing >50% low-quality ($Q \leq 5$) bases from raw data. All of the downstream analyses were based on clean data with high quality. For read mapping, the reference genome (Macaca_mulatta, Mmul_10, release-104) were downloaded from the Ensembl website. An index of the reference genome was built using Hisat2 v2.2.1, and paired-end clean reads were aligned to the reference genome using Hisat2 v2.2.1[53]. Gene expression level and the transcription level were calculated by HTSeq v1.99.2 and edgeR v3.36.0 to screen differentially expressed genes between groups[53–55]. A Gene Ontology (GO) enrichment analysis and Kyoto Encyclopedia of Genes and Genomes (KEGG) enrichment analysis of upregulated and downregulated genes was implemented by cluster Profiler R package v4.2.2[56]. A ggplot2 R package v3.4.0 is used to draw bar charts, pie charts, and violin charts[57].

## Tissue dissociation and preparation of single-cell suspensions

Testes from ZIKV-infected A6 mice were harvested at 14 dpi. Testes from uninfected mice served as controls. Tissues were dissociated into single cells in dissociation solution (0.35% collagenase IV5, 2 mg/ml papain, 120 Units/ml DNase I) in 37 °C water bath with shaking for 20 min at 100 rpm. Digestion was terminated with 1× PBS containing 10% fetal bovine serum (FBS, V/V), then pipetting 5–10 times with a Pasteur pipette. The resulting cell suspension was filtered by passing through 70–30 μm stacked cell strainer and centrifuged at $300 \times g$ for 5 min at 4 °C. The cell pellet was resuspended in 100 μl 1× PBS (0.04% BSA) and added with 1 ml 1× red blood cell lysis buffer (MACS 130-094-183, 10×) and incubated at room temperature or on ice for 2–10 min to lyse remaining red blood cells. After incubation, the suspension was centrifuged at 300 × g for 5 min at room temperature. The suspension was resuspended in 100 μl Dead Cell Removal MicroBeads (MACS 130-090-101) and remove dead cells using Miltenyi ® Dead Cell Removal Kit (MACS 130-090-101). Then the suspension was resuspended in 1× PBS (0.04% BSA) and centrifuged at $300 \times g$ for 3 min at 4 °C (repeat twice). The cell pellet was resuspended in 50 μl of 1× PBS (0.04% BSA). The overall cell viability was confirmed by trypan blue exclusion, which needed to be above 85%, single cell suspensions were counted using a haemocytometer/Countess II Automated Cell Counter, and concentration adjusted to 700–1200 cells/μl.

## Chromium 10× Genomics library and sequencing

Single-cell suspensions were loaded to 10× Chromium to capture single cells according to the manufacturer's instructions of 10× Genomics Chromium Single-Cell 3′ kit (V3). The following cDNA amplification and library construction steps were performed according to the standard protocol. Libraries were sequenced on an Illumina NextSeq500 sequencing system (paired-end multiplexing run, 150 bp) by LC-Bio Technology co. ltd., (Hang Zhou, China).

## Sc RNA-seq analysis

Sequencing results were demultiplexed and converted to FASTQ format using Illumina bcl2fastq software. Sample demultiplexing, barcode processing, and single-cell 3′ gene counting by using the Cell Ranger v5.0.1[58] and scRNA-seq data were aligned to Ensembl genome GRCm38 reference genome (release-95). The Cell Ranger output was loaded into Seurat v4.1.1 be used to Dimensional reduction, clustering, and analysis of scRNA-seq data[59]. Overall, 22,988 cells passed the quality control threshold: all genes expressed in less than one cells were removed, number of genes expressed per cell >500 as low cutoff, the percent of mitochondrial-DNA derived gene-expression <25%.

To visualize the data, we further reduced the dimensionality of all 22,988 cells using Seurat and used t-SNE to project the cells into 2D space, The steps include: 1. Using the LogNormalize method of the "Normalization" function of the Seurat software to calculated the expression value of genes; 2. PCA (Principal component analysis) analysis was performed using the normalized expression value. Within all the PCs, the top 10 PCs were used to do clustering and t-SNE analysis; 3. To find clusters, selecting weighted Shared Nearest Neighbor (SNN) graph-based clustering method. Marker genes for each cluster were identified with the Bimod Likelihood-ratio test with default parameters via the FindAllMarkers function in Seurat. This selects markers genes which are expressed in more than 10% of the cells in a cluster and average log (Fold Change) of greater than 0.25. Expression of upregulated and downregulated genes analyzed by Kyoto Encyclopedia of Genes and Genomes (KEGG) enrichment analysis was conducted by cluster Profiler R package v4.2.2[56]. Intercellular communication in testes was analysis by using CellChat v1.4.0 as previously reported[8].

## Statistics and reproducibility

The images of IFA, IHC, HE staining, and trypan blue staining were performed in at least three independent experiments. All statistical analyses were performed using SPSS 17.0 Software (IBM, Armonk, NY, USA) and rechecked by Microsoft Excel 2016. The quantitative data between two groups with normal distributions were analyzed using the two-sided Student's $t$ test. Data with abnormal distributions of variance were analyzed by nonparametric Mann–Whitney $U$ test. Changes in body weight and symptom score were analyzed using repeated-measures ANOVA. The relative titer of anti-ZIKV antibodies was analyzed using the one-way ANOVA. All results were presented as the mean ± standard error of the mean (SEM) in this research from at least three repeats. $P < 0.05$ was considered as has statistically significant between two groups. Correlation between expression level of C1q genes and S100A4 gene was analyzed by regression analysis.

## Reporting summary

Further information on research design is available in the Nature Portfolio Reporting Summary linked to this article.

## Data availability

All data needed to evaluate the conclusions in the paper are present in the paper and/or the Supplementary Materials. For read mapping of the RNA-Seq data from *macaca leonine* testes, the reference genome (Macaca_mulatta, Mmul_10, release-104) was downloaded from the Ensembl website. scRNA-seq data were aligned to Ensembl genome

GRCm38 reference genome (release-95). The raw scRNA-seq data generated in this study have been deposited in the NCBI Sequence Read Archive under accession PRJNA756783. The raw RNA-seq data generated in this study have been deposited in the NCBI Sequence Read Archive under accession PRJNA756717. Source data are provided in this paper.

## Code availability

All code associated with this manuscript has been uploaded to GitHub (https://github.com/anlab2022/ZIKV-testis-sc-data).

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

## Acknowledgements

We appreciate the grants from the National Key Research and Development Plan of China (2021YFC2300202 to J.A.), the National Natural Science Foundation of China (U1902210 to J.A., 81871641 to P.G.W., 81972979 to J.A., 81902048 to Z.Y.S., 82172266 to P.G.W.), and the Support Project of High-level Teachers in Beijing Municipal Universities in the Period of 13th Five-year Plan (IDHT20190510) to J.A. The funders had no role in study design, data collection, and analysis, decision to publish, or preparation of the manuscript. We are also grateful to Professor Zhi-Hai Qin (Institute of Biophysics, Chinese Academy of Science) for his constructive advice on research design and refinement.

## Author contributions

W. Yang, L.B. Liu, and Y.H. Wu designed and performed the experiments, and analyzed the data. F.L. Liu and Y.T. Zheng designed and performed the experiments with northern pigtailed macaques. Z.D. Zhen and Z.Y. Sheng helped to design the experiments. Z.R. Song and J.T. Chang helped to perform some animal experiments. D.Y. Fan helped to maintain cells, viruses, and reagents. J. An and P.G. Wang conceived the project, analyzed the data, wrote and reviewed the manuscript. J. An coordinated the whole research.

## Competing interests

The authors declare no competing interests.
