## [Peer Review File · Nature Communications]

Reviewer comments, first round review

Reviewer #1 (Remarks to the Author):

This is a follow-up study focusing on the mechanism of the ZKV effect on spermatogenic cells in three species. These are my comments and concerns:

The strengths of the study:

- 1) Use of a relevant non-human primate (NHP) model (Pigtail macaque) in some of the experiments for addressing the hypothesis
- 2) Use of single-cell Seq analysis to understand the cellular response to Zika in the testes

The Concerns:

- 1) The mouse, monkey, and human cell lines data are confusing. The results should be re-organized to be reported separately for each species and be easy to follow for the readers. This is very important, as A6 mice (IfnaR deficient) are severely immunocompromised and exhibits very severe damage of testes that do not represent human features of Zika virus testicular infection. ZIKV-infected and recovered males do not show any of the features of testicular dysfunction seen in A6 mice. The limitations of mouse models need to be addressed clearly in the discussion.
- 2) The mouse testes organoid is not well characterized. More data is appreciated to see how much mimicking in vivo condition.
- 3) The single-cell RNA Seq experiments conducted in the NHP model instead of A6 mice would give more meaningful data.
- 4) Most of the experiments are conducted in A6 mice where infiltration of S100A4+ macrophages was shown in their previous study. The immune-compromised nature of these mice is the main reason for such massive infiltration in the testis that is not seen in ZIKV-infected human testes.
- 5) The analysis of single-cell data is very vague. The clusters are generated based on the host response instead of clustering the data into different cell types using cell-specific markers. It is not clear which cell types are present in each cluster. Ex; Cluster 0, 7, and 22 have macrophages- are they infected? Which of these clusters have ZIKV-infected cells? The data should be first clustered in different cell types, percent of infiltrated macrophages and other immune cells should be analyzed, the cell abundance ratio of different cell types including loss of spermatogonia; and then GSEA analysis should be conducted to analyze cell type-specific response. The methodology is not transparent- were the infected cells fixed before generating the library?
- 6) The data from the drug treatment experiments need to be repeated in the pigtail macaques (future studies) to strengthen the fact that blocking S100A4 cells and C1q and C3 expression in testes can reverse pathology and fertility loss.

Overall, the A6 mice data does not add any significant insights into the human testicular infection of Zika, and more experiments should be conducted to characterize these mechanisms in macaque or immunocompetent mouse models (hSTAT knock-in in BL6 background).

Reviewer #2 (Remarks to the Author):

This manuscript by Yang and colleagues reports an investigation into the role of S100A4+ macrophage in the immunological basis of spermatogenic cell damage in Zika virus (ZIKV) infected male mice. The authors followed up on their previous publication in PLoS Pathogens to show here that S100A4+ macrophage infiltrates the testes and the undergo further transcriptional changes to express mediators of classical complement. They showed that, both in mice that lacks S100A4 and

those treated with S100A4 inhibitor (niclosamide), testicular damage was reduced. They concluded that targeting S100A4 could be a therapeutic approach to preventing infertility that could be associated with ZIKV infection.

This is an interesting study from an immunological perspective. However, there is a lack of detailed investigation into ZIKV infection, which confounds the interpretation and conclusion of this study. My major concerns are as follows:

1. The main problem with this manuscript is the assumption that the macrophages were responding to spermatogenic cell infection and hence caused the damage. There is an alternative explanation – the macrophages that migrated into the testes could become infected with ZIKV and thus amplified the infection. The resultant innate immune response, including complement activation could thus have resulted in the observed immunopathological changes. The following comments are related to this main concern and provide granularity on where the data is deficient.

2. Lines 38-51 and Figure 1. The authors performed RNAseq on the testes of ZIKV infected animals. Could this data provide indication on what cells were infected with ZIKV? Infected macrophages would fundamentally alter the narrative and conclusion of this paper.

3. Lines 57-63 and Extended Data Figure 3a-b. The immunofluorescence micrograph of the organoid is unclear. The use of confocal microscopy with z-stacking would have been more appropriate in this instance. As it is, it is impossible to tell if the fluorescent signals were true co-localization or just one stacked on top of another.

4. Lines 57-63 and Extended Data Figure 3c-d. The co-localization signal would have benefited from a quantitative analysis. Unlike what was described in the text, the ZIKV and caspase-3 signals in these figure panels, to this reviewer, do appear to be co-localized.

5. Lines 120-130 and Figure 2k-m. The notion that activation of the classical complement pathway is critical for ZIKV infected spermatogenic cell death is not entirely supported by this experiment. This is especially since the data in Extended Data Figure 3 is not entirely convincing. The authors should consider more objective ways to assess cell death than using trypan blue.

6. Lines 132-161 and Figure 3. The lack of probing for ZIKV proteins in these analyses make data interpretation problematic. The authors suggest from this set of data that the C1q signal colocalization with macrophage markers indicate the source of C1q. A plausible alternative explanation is that C1q is activated because these macrophages were infected with ZIKV. The use of SA6 mice cannot exclude this possibility as the lack of testicular damage could again be explained by the lack of infection amplification from macrophage infiltration. Furthermore, the authors had not assessed if ZIKV infection in SA6 mice produced equivalent levels of viremia and viral dissemination as A6 mice.

7. Lines 173-193. The niclosamide data is interesting but at the same time problematic. There is no explanation on why sulindac, also a S100A4 inhibitor did not show any therapeutic effect. More importantly, it is unclear from the data if by blocking S100A4, there was a lesser extent of macrophage infiltration of the testes and hence reduced ZIKV amplification – there was reduced ZIKV infection load in the testes after all (Extended Data Figure 9h).

8. Finally, the manuscript could also benefit from careful editing of typographical errors and to provide more accurate descriptions of both data and concepts.

Reviewer #3 (Remarks to the Author):

The authors follow up on their previous study and propose that the S100A4+ macrophages are primarily involved in the induction of the classical complement pathway that is responsible for spermatogenic cell death in Zika virus infection. To demonstrate this, they showed that there was an influx of immune cells in the testes of ZIKV-infected A6 mice using scRNA-Seq. They also found

that the genes for the classical complement pathway were upregulated while those of others such as apoptosis were not which was confirmed by IFA, ELISA, and RT-qPCR in various experiments using organoids from A6 and Balb/c mice, JKT-1 cells, primary testicular cells, and testes from infected northern pigtailed macaques. Furthermore, an in vitro assay using anti-ZIKV antibodies with JKT-1 showed that the activated complement system but not the infection was primarily responsible for cell death. IFA was used to show that C1q co-localized with macrophage and myeloid markers as well as S100A4. The authors also performed studies using S100A4 deficient mice and found that these mice had less inflammation and more intact seminiferous tubules. Finally, niclosamide, an S100A4 inhibitor, was found to prevent testicular damage although it did not have any effect on the infection and also improved fertility compared to mock-treated mice. This is a valuable study as it identifies the mechanism as well as describes a potential therapeutic that will have broader implications towards understanding viral pathogenesis. The study is well designed and the conclusions are supported by the evidence. Following are some concerns:

Major concerns:

- There is a very limited description of how the transcriptomic analysis was performed. In order to reproduce the analysis, a more detailed methodology will be required.
- There is no description of how the cells were annotated. It has been proposed that clusters 0 and 22 are macrophages that differentiated from cluster 7 monocytes. It will be very helpful to see how all the different cell types were defined. It may also be worthwhile to subset these macrophage/monocyte clusters and analyze them separately from other cell types.
- Since the scRNA-Seq data is based on multiple subjects, it will be helpful to show t-SNE colored by individuals. Similarly, the proportions of cell types by individual will be helpful.
- Fig 3C shows that S100A4 is expressed in the T cell cluster as well. It will be helpful if the authors can comment on that.
- Line 67: most of them were expressed by monocytes or macrophages in clusters 0, 5, 7, and 22
- Cluster 5 is primarily composed of Fibroblasts. Are fibroblasts also involved in the complement activation?

Minor concerns:

- Panel 1c uses a different t-SNE compared to d-i.
- Providing the results of differential gene expression for Fig 1k-l with p-values as Supplementary Table.
- It has been mentioned that the ggplot2 package was used for image processing, base recognition and KEGG enrichment analysis. This needs to be confirmed.
- The processed data should be uploaded to a public repository such as NCBI GEO.

Response to Reviewers' comments, from Authors

Reviewer #1 (Remarks to the Author):

This is a follow-up study focusing on the mechanism of the ZKV effect on spermatogenic cells in three species. These are my comments and concerns:

The strengths of the study:

- 1) Use of a relevant non-human primate (NHP) model (Pigtail macaque) in some of the experiments for addressing the hypothesis
- 2) Use of single-cell Seq analysis to understand the cellular response to Zika in the testes

Reply: Thank you for the encouraging comments and paying constant attention on our work. In the past months, we re-wrote the manuscript based on our recent progress and the Reviewer's comments from last submission. Comparing to old version, the new manuscript includes (1) the influence of ZIKV infection on the development and differentiation of spermatogenic cells, (2) the influence of ZIKV infection on the function of monocyte/macrophage, (3) the effects of complement inhibitor C1INH on ZIKV infection, (4) and results from the experiments suggested by three reviewers in last submission. Hope all the revision can address your concern. Here is the point-to-point replies.

The Concerns:

1) The mouse, monkey, and human cell lines data are confusing. The results should be re-organized to be reported separately for each species and be easy to follow for the readers. This is very important, as A6 mice (IfnaR deficient) are severely immunocompromised and exhibits very severe damage of testes that do not represent human features of Zika virus testicular infection. ZIKV-infected and recovered males do not show any of the features of testicular dysfunction seen in A6 mice. The limitations of mouse models need to be addressed clearly in the discussion.

Reply: Thank you for your suggestion. We have re-organized the Results sections to separate the data for each species. A majority of A6 mouse experiments are placed at the beginning of the Results section at **lines 81-264**, followed by the macaque experiments at **lines 266-309**, and finally the treatment experiments of A6 mice at **lines 311-354**.

Regarding the pathological changes in ZIKV-infected testis, the representativeness of IFNAR knockout mouse is a reasonable concern. As clinical evidences show, although oligospermia and hemospermia have been well-documented¹⁻³, and infectious particles or viral RNA are detected in the semen of ZIKV patients for a long time^{4,5}, the severe symptom such as orchitis is rarely reported. In this research, we used northern pigtailed macaques to address the concern, because nonhuman primates can better mimic the

pathogenesis of patients with ZIKV infection. As revealed by IFA and RNA sequencing, although the testicular pathological changes are mild in ZIKV-infected macaques, they did show the characteristics similar to those observed in A6 mice, including the infiltration of S100A4+ monocyte/macrophage (Fig. 6d), the complement activation (Fig. 6d,g & j), and the damage of spermatogenic cells (Fig. 6c,d,f & j). These results suggest that the lesions observed in A6 mice are similar to non-human primate. Nevertheless, in the presence of intact IFN- α/β signaling pathway, macaques had much stronger responses regarding the cell junction, cell survival and tissue repair (Supplementary Fig. S8b), which effectively restricted the damage to spermatogenic cells, and make the pathological change in the testis much milder than A6 mice.

The detailed description was added as a new paragraph in revised manuscript (line 411-426).

2) The mouse testes organoid is not well characterized. More data is appreciated to see how much mimicking in vivo condition.

Reply: Thanks for your valuable counsel. As results from testicular organoid are not important for the manuscript and need further characterization, the results relating to testicular organoid experiments were removed from the manuscript. This modification had no effect on our conclusions.

3) The single-cell RNA Seq experiments conducted in the NHP model instead of A6 mice would give more meaningful data.

Reply: We agree with you that the single-cell RNA Seq experiments conducted in the NHP model instead of A6 mice would give more meaningful data NHP model. However, this doesn't mean the results from mice are nonsense. This manuscript is a follow-up study of our previous work using A6 and SA6 mice⁶, so that the single-cell RNA sequencing using these mice is a reasonable and intermediate process from mice to NHP. Moreover, the sequencing results from mice are easily to validated by further experiments, just as what we presented in the manuscript. This is a common strategy in virology research. It will be thought unethical to perform experiments directly using NHP without solid data from small animals, and will not be approved by the ethics committee. That's why we chose mice rather than NHP for the single-cell RNA sequencing.

Moreover, our results showed that there were indeed some similarities of pathological changes in testes between A6 mouse and northern pigtailed macaques, including decreased numbers of spermatogenic cells (Fig. 6e), the infiltration of S100A4+ macrophages (Fig. 6d) and activation of complement (Fig. 6d,g & j), although the extent was much slight in the northern pigtailed macaques. These data implied that the sequencing results from mice were valuable in our study.

4) Most of the experiments are conducted in A6 mice where infiltration of S100A4+

macrophages was shown in their previous study. The immune-compromised nature of these mice is the main reason for such massive infiltration in the testis that is not seen in ZIKV-infected human testes.

Reply: Yes, as you pointed out, there is no report regarding S100A4+ macrophage infiltration in ZIKV-infected human testes. However, CD68+/CD163+ testicular macrophages isolated from uninfected human testicular tissue had been proved to support ZIKV infection *in vivo*⁷. In addition, in testes of ZIKV-infected baboons, macrophages were observed to invade into the interior of seminiferous tubules⁸. These results suggest a possibility that monocytes/macrophages are likely to invade into the interior of seminiferous tubules in human testes after ZIKV infection. In line with these results, S100A4+ monocytes/macrophages infiltration was also detected in testis from ZIKV-infected northern pigtailed macaques in our investigation (**Fig. 6d**). Thus, we consider that the results conducted in A6 mice can reflect the mechanism underlying ZIKV-induced testes damage in humans to some extent.

5) The analysis of single-cell data is very vague. The clusters are generated based on the host response instead of clustering the data into different cell types using cell-specific markers. It is not clear which cell types are present in each cluster. Ex; Cluster 0, 7, and 22 have macrophages- are they infected? Which of these clusters have ZIKV-infected cells? The data should be first clustered in different cell types, percent of infiltrated macrophages and other immune cells should be analyzed, the cell abundance ratio of different cell types including loss of spermatogonia; and then GSEA analysis should be conducted to analyze cell type-specific response. The methodology is not transparent- were the infected cells fixed before generating the library?

Reply: We appreciate for your valuable comments. According to your suggestion, we clustered single-cell RNA-Seq data into different cell types by using cell-specific markers^{9,10}. All testicular cells were preliminarily divided into nine cell clusters, including exogenic monocytes/macrophages, testicular macrophages, Sertoli cells (suspected), Leydig cells, spermatogenic cells, T/NK cells, fibroblasts, endothelial cells and granulocytes (**line 82-89**) (see **Fig. 1a** and **Supplementary Fig. S1a** for details).

Next, we analyzed the changes in the number and proportion of testicular cells (**Fig. 1b** and **Supplementary Fig. S1b**), and found that the number of spermatogenic cells decreased significantly after infection, while the number of immune cells increased significantly (**lines 90-93**).

To clarify the composition of spermatogenic cells, we further clustered them into spermatids, spermatocyte and spermatogonia (**Fig. 2a, b**), and observed that ZIKV infection affected the differentiation and development of spermatogonia (**Fig. 2d, e**). However, ZIKV infection did not lead to the high expression of genes related to multiple death pathways of these spermatogenic cells, indicating that ZIKV infection is not the immediate cause of spermatogenic cells damage (**lines 108-121**), which is also consistent with the results reported in existing

studies¹¹.

Most of exogenous monocytes and macrophages were infected by ZIKV (**Fig. 1c and Fig. 3f**), and ZIKV infection would up-regulate complement system related genes in these cells (**Fig. 3g, lines 99-101 and 165-183**). The GSEA analysis results of KEGG pathways also show that exogenous monocytes and macrophages have similar phagocytic function and inflammatory response (**Supplementary Fig. S4c**), and their KEGG enrichment pathways are basically the same, indicating that these two cell clusters have homology (**lines 156-163**).

The detailed method of single-cell RNA-Seq is supplemented in Materials and Methods section (**lines 608-683**).

6) The data from the drug treatment experiments need to be repeated in the pigtail macaques (future studies) to strengthen the fact that blocking S100A4 cells and C1q and C3 expression in testes can reverse pathology and fertility loss.

Reply: Thanks for your valuable counsel. As you proposed for future studies, we are planning to test the effects of S100A4 inhibitor on the pigtail macaques in the future, and we have gotten a funding for the research-- “The characteristics and mechanism underlying testis damage induced by ZIKV infection in northern pig-tailed macaques, Reference number: U1902210”. However, because most NHP animals were used for COVID-19 study in these years, this experiment can’t be done at the moment. With the fade of COVID-19 pandemic, we hope to perform the above project in the future, and we would be happy to share our data with researchers in the field at that time.

Overall, the A6 mice data does not add any significant insights into the human testicular infection of Zika, and more experiments should be conducted to characterize these mechanisms in macaque or immunocompetent mouse models (hSTAT knock-in in BL6 background).

Reply: Thank you for your suggestion on hSTAT knock-in mice. In our last work in 2020, we have received similar comments as yours to repeat the results using hSTAT2 knock-in mice. We asked Professor Michael Diamond for help and then ordered the mice from Jackson Lab. A pair of them arrived in Beijing last month—it took longer time than usual to delivery for Jackson lab to China. The experiments with these mice are expected to be initiated this year.

Although we are going to test the results on hSTAT2 mice this year and on NHP as soon as the animals are available, we can’t agree with you on the significance of A6 mice for the study of ZIKV infection. Since 2016, most progress on ZIKV have been obtained with the help of A6 or A129 mice. Human STAT2 knock-in mice have advantage over A6 mice, but they are used by a few labs and do not replace A6 mice, implying that A6 mice also have their own advantage. Regarding NHP, they are important, but they can be used in most experiments in the field on vector-borne viruses both economically and ethically. In fact, mice are still the dominant animal model in the study on ZIKV infection.

Reviewer #2 (Remarks to the Author):

This manuscript by Yang and colleagues reports an investigation into the role of S100A4+ macrophage in the immunological basis of spermatogenic cell damage in Zika virus (ZIKV) infected male mice. The authors followed up on their previous publication in PLoS Pathogens to show here that S100A4+ macrophage infiltrates the testes and the undergo further transcriptional changes to express mediators of classical complement. They showed that, both in mice that lacks S100A4 and those treated with S100A4 inhibitor (niclosamide), testicular damage was reduced. They concluded that targeting S100A4 could be a therapeutic approach to preventing infertility that could be associated with ZIKV infection.

Reply: Thank you for your positive comments and specific suggestions. To improve the manuscript, we revised the grammar and syntax throughout the text and reanalyzed the data of single-cell RNA Seq. The other comments are replied to point-to-point below.

This is an interesting study from an immunological perspective. However, there is a lack of detailed investigation into ZIKV infection, which confounds the interpretation and conclusion of this study. My major concerns are as follows:

1. The main problem with this manuscript is the assumption that the macrophages were responding to spermatogenic cell infection and hence caused the damage. There is an alternative explanation – the macrophages that migrated into the testes could become infected with ZIKV and thus amplified the infection. The resultant innate immune response, including complement activation could thus have resulted in the observed immunopathological changes. The following comments are related to this main concern and provide granularity on where the data is deficient.

Reply: Yes, as you pointed out, the macrophage does play important roles in ZIKV infection in the testis.

The current manuscript is a follow-up study in our lab since the pandemic of ZIKV in 2016. In one paper in 2017¹², we identified Sertoli cells and macrophage in testis were the main target cells of ZIKV in testes. In another paper in 2020⁶, we carefully characterized the macrophage (mainly S100A4+) appeared in ZIKV-infected testes and identified their contribution to ZIKV infection in testes. The conclusion is similar to the explanation you proposed. However, in these works, we didn't find out the mechanisms underlying the damage of spermatogenic cell – we had hypothesized that the spermatogenic cell were attacked by macrophage or CD8+ T cell but finally found the hypothesis was not right. From then on, we performed single-cell RNA sequencing and other experiments, and noted that the complement activation is the main reason for the loss of spermatogenic cell. These results are presented in the current manuscript. The explanation you proposed briefly and accurately summarized our previous work and current work.

The following comments will be replied point-to-point.

2. Lines 38-51 and Figure 1. The authors performed RNAseq on the testes of ZIKV infected animals. Could this data provide indication on what cells were infected with ZIKV? Infected macrophages would fundamentally alter the narrative and conclusion of this paper.

Reply: Thanks for your comments. Based on the cell-specific markers^{9,10}, we have preliminarily clustered all testicular cells into nine cell clusters, including exogenic monocytes/macrophages, testicular macrophages, Sertoli cells (suspected), Leydig cells, spermatogenic cells, T/NK cells, fibroblasts, endothelial cells and granulocytes (see Fig. 1a and Supplementary Fig. S1a for details), the relevant results are in lines 82-89.

As expected, ZIKV RNA was detected only in ZIKV-infected testes. Most cells had a substantial expression of ZIKV RNA, especially monocytes and macrophages (Fig. 1c and in lines 99-100), suggesting their susceptibility to ZIKV infection. To investigate the effect of ZIKV infection on spermatogenic cells, we further divided the spermatogenic cell cluster into spermatids, spermatocyte and spermatogonia (Fig. 2a, b), and found that ZIKV infection affected the differentiation and development of spermatogonia (Fig. 2d, e). However, ZIKV infection did not lead to the high expression of genes related to multiple death pathways of these spermatogenic cells, indicating that ZIKV infection is not the immediate cause of spermatogenic cells damage, which is also consistent with the results reported in existing studies¹¹. The corresponding results are in lines 108-147.

Most of exogenous monocytes and macrophages were infected by ZIKV (Fig. 1c and Fig. 3f), and ZIKV infection would lead to the up-regulation of complement system related genes in these cells (Fig. 3g).

3. Lines 57-63 and Extended Data Figure 3a-b. The immunofluorescence micrograph of the organoid is unclear. The use of confocal microscopy with z-stacking would have been more appropriate in this instance. As it is, it is impossible to tell if the fluorescent signals were true co-localization or just one stacked on top of another.

Reply: As results from testicular organoid are not important for the manuscript and need further characterization, these results were removed from the revised manuscript.

4. Lines 57-63 and Extended Data Figure 3c-d. The co-localization signal would have benefited from a quantitative analysis. Unlike what was described in the text, the ZIKV and caspase-3 signals in these figure panels, to this reviewer, do appear to be co-localized.

Reply: We are sorry for the confusing description. By these supplementary results, we wanted to compare the expression of caspase-3 et al, rather than their co-localization.

Regarding their co-localization, as you pointed out, there is really a small amount of DDX4 co-located with Caspase-3 signal (Supplementary Fig. S2c, d in

revised manuscript). The detection of caspase in DDX4 cells were not surprised, because these cells naturally had slightly apoptosis. Moreover, ZIKV is able to induce apoptosis in infected cells.

As to the expression level the caspase genes, which were the aim of this experiment, ZIKV-infected samples showed no increase of just slight increase. Our in-vitro results showed similar results in primary testicular cells or JKT-1 cells (**Supplementary Fig. S2c, d** and in **lines 143-145**). Moreover, single-cell RNA-seq data displayed that multiple death pathway-related genes are not highly expressed in ZIKV-infected spermatogenic cells (**Fig. 2h, lines 134-143**). Based on the above results, we made the statement that ZIKV infection dose not up-regulate Caspase expression.

In this version, to avoid the confusion, we have revised the description in our manuscript accordingly. We also accepted your comments to count the number and the percentage of Caspase-3 positive cells in JKT-1 cells (**See the figure below for details, a, b**) and primary testicular cells (**See the figure below for details, c, d**), no significant difference was detected in ZIKV-infected cells as compared with control cells.

5. Lines 120-130 and Figure 2k-m. The notion that activation of the classical complement pathway is critical for ZIKV infected spermatogenic cell death is not entirely supported by this experiment. This is especially since the data in Extended Data Figure 3 is not entirely convincing. The authors should consider more objective ways to assess cell death than using trypan blue.

Reply: Thank you for the comments. To validate the role of complement activation, we have designed and performed the following experiments: First, we

used intracellular LDH as an alternative to trypan blue to quantify the damage of the complement system to JKT-1 cells (Fig. 4k). The results showed that the concentration of intracellular LDH significantly decreased in the presence of both ZIKV and anti-ZIKV serum at 72-hour post infection, indicating that the complement system did damage ZIKV-infected JKT-1 cells (Fig. 4l, m). The corresponding description can be found in lines 215-228. Second, we treated ZIKV-infected A6 male mice with C1INH, a classical complement activation pathway inhibitor. The results displayed that C1INH could alleviate the ZIKV-induced pathological damage of testis and reduce the testicular atrophy and inflammatory reaction (Supplementary Fig. S9a-c). More importantly, the number of spermatogenic cells co-stained with MAC in the testis of C1INH treated mice decreased significantly (Supplementary Fig. S9d, e), indicating that C1INH can alleviate testicular pathological damage by inhibiting the activation of complement system. However, the viral load in testis and blood of C1INH-treated mice was significantly higher (Supplementary Fig. S9f, g), and the disease of C1INH-treated mice was more serious (Supplementary Fig. S9h). These results suggested that C1INH could alleviate the pathogenic damage in ZIKV-infected testes but could not inhibit the replication of ZIKV in the whole body. Taken together, the above results suggested that activation of the classical complement pathway, not ZIKV infection itself, is critical for ZIKV infected spermatogenic cell death. The corresponding description is in lines 312-326.

6. Lines 132-161 and Figure 3. The lack of probing for ZIKV proteins in these analyses make data interpretation problematic. The authors suggest from this set of data that the C1q signal colocalization with macrophage markers indicate the source of C1q. A plausible alternative explanation is that C1q is activated because these macrophages were infected with ZIKV. The use of SA6 mice cannot exclude this possibility as the lack of testicular damage could again be explained by the lack of infection amplification from macrophage infiltration. Furthermore, the authors had not assessed if ZIKV infection in SA6 mice produced equivalent levels of viremia and viral dissemination as A6 mice.

Reply: Thank you for your comments. By thinking about your questions, we further analyzed the effect of ZIKV infection on the expression of complement system genes of monocytes/macrophages. Based on results from single-cell RNA-Seq data, C1q and C3 were expressed in all exogenous monocytes/macrophages, and their expression levels did increase after ZIKV infection (Fig. 3g). The corresponding description is added in lines 179-183. More importantly, in ZIKV-infected testicular tissues, C1q is mainly expressed by exogenous monocytes/macrophages, which supports the key role of monocytes/macrophages in the activation of testicular complement system.

In addition, in our previous study⁶, we evaluated the viremia in SA6 mice and A6, and viral loads in blood of SA6 mice were generally lower than that in A6 mice (See the figure below for details, a). As to viral dissemination, we detected viral load in testis from both kinds of mice, and we found different dynamics of ZIKV

RNA in the testes in case of the absence of S100A4+ macrophages: in SA6 mice, testicular viral load at 7 dpi was higher than that at 14 dpi (See the figure below for details, c), whereas in A6 mice the testicular viral load at 7 dpi was lower than that at 14 dpi (See the figure below for details, d). These results suggested that bone marrow-derived S100A4+ macrophages were not required for ZIKV replication in the testes at the early stage of ZIKV infection, but they played an important role in testis infection after the acute phase of ZIKV infection.

In C1INH-treated ZIKV-infected A6 mice, we considered that the testicular viral load is not directly related to testicular pathological damage (reply 5 above) (Supplementary Fig. S9). Therefore, the different degree of testicular complement system activation between SA6 mice and A6 mice is the major factor leading to the different level of testicular pathological damage.

7. Lines 173-193. The niclosamide data is interesting but at the same time problematic.

There is no explanation on why sulindac, also a S100A4 inhibitor did not show any therapeutic effect. More importantly, it is unclear from the data if by blocking S100A4, there was a lesser extent of macrophage infiltration of the testes and hence reduced ZIKV amplification – there was reduced ZIKV infection load in the testes after all (Extended Data Figure 9h).

Reply: Thank you very much for your comments. In the Discussion section, we have given the explanation regarding the mechanism of sulindac and niclosamide inhibiting S100A4. Sulindac interdict S100A4 synthesis by intervening in β -catenin signaling¹³, but it cannot block the promotion effect of other cytokines on the synthesis of S100A4 protein such as fibroblast growth factor-2 (FGF-2) and transforming growth factor-beta1 (TGF-beta1)^{14,15}. While the inhibitory effect of niclosamide on S100A4 expression may be caused by blocking the S100A4 gene promoter, as S100A4 expression didn't decrease in the colorectal cancer (CRC) cell line with S100A4 overexpression driven by CMV promoter after niclosamide treatment¹⁶, which might be the reason for the different therapeutic effect between these two drugs in inhibiting S100A4 expression.

Actually, niclosamide did repressed the S100A4+ monocytes/macrophages infiltration in testes from ZIKV-infected A6 mice (**Supplementary Fig. S10a**). And as we had proved that these monocytes/macrophages could support ZIKV infection and replication in testes⁶, lesser S100A4+ monocytes/macrophages infiltration in testes from niclosamide-treated mice did result in lower testicular viral load (**Supplementary Fig. S10b**).

8. Finally, the manuscript could also benefit from careful editing of typographical errors and to provide more accurate descriptions of both data and concepts.

Reply: We are very sorry for the errors in English grammar and spelling in this manuscript. We have asked an English language editing service (American Journal Expert, AJE) for revising our description and we hope it can meet the requirements of the Journal.

Reviewer #3 (Remarks to the Author):

The authors follow up on their previous study and propose that the S100A4+ macrophages are primarily involved in the induction of the classical complement pathway that is responsible for spermatogenic cell death in Zika virus infection. To demonstrate this, they showed that there was an influx of immune cells in the testes of ZIKV-infected A6 mice using scRNA-Seq. They also found that the genes for the classical complement pathway were upregulated while those of others such as apoptosis were not which was confirmed by IFA, ELISA, and RT-qPCR in various experiments using organoids from A6 and Balb/c mice, JKT-1 cells, primary testicular cells, and testes from infected northern pigtailed macaques. Furthermore, an in vitro assay using anti-ZIKV antibodies with JKT-1 showed that the activated complement system but not the infection was primarily responsible for cell death. IFA was used to show that C1q co-localized with macrophage and myeloid markers as well as S100A4. The authors

also performed studies using S100A4 deficient mice and found that these mice had less inflammation and more intact seminiferous tubules. Finally, niclosamide, an S100A4 inhibitor, was found to prevent testicular damage although it did not have any effect on the infection and also improved fertility compared to mock-treated mice.

This is a valuable study as it identifies the mechanism as well as describes a potential therapeutic that will have broader implications towards understanding viral pathogenesis. The study is well designed and the conclusions are supported by the evidence. Following are some concerns:

Reply: Thank you very much for your encouragement. We have replied to your questions and hope to meet your requirements.

Major concerns:

- There is a very limited description of how the transcriptomic analysis was performed. In order to reproduce the analysis, a more detailed methodology will be required.

Reply: According to your suggestion, the detailed method of transcriptomic analysis of testicular tissue from macaques and single-cell RNA-Seq analysis of testes from A6 mice is supplemented in Materials and Methods section, including the library preparation and brief protocol of RNA sequencing, and preparation of single-cell suspensions, chromium 10× Genomics library and brief protocol of single-cell RNA-Seq. Please see lines 613-688 for details.

There is no description of how the cells were annotated. It has been proposed that clusters 0 and 22 are macrophages that differentiated from cluster 7 monocytes. It will be very helpful to see how all the different cell types were defined. It may also be worthwhile to subset these macrophage/monocyte clusters and analyze them separately from other cell types.

Reply: Thank you for your significant reminding. After reanalyzed the data of single-cell RNA Seq, we make a detailed description of the characterization of

different cell clusters in Results section in **lines 82-97**. In brief, all cells in single-cell RNA-Seq was defined based on the cell-specific markers of testicular cells and immune cells^{9,10}, and testicular cells were preliminarily clustered into nine cell types, including exogenic monocytes/macrophages, testicular macrophages, Sertoli cells (suspected), Leydig cells, spermatogenic cells, T/NK cells, fibroblasts, endothelial cells and granulocytes (**see Fig. 1a and Supplementary Fig. S1a for details**), the relevant results are in **lines 82-89**.

In order to separate from other cells, exogenic monocytes/macrophages were clustered based on their cell-specific markers, including Cd38, Cd68, Itgam, Cd14, S100a4, Marco, Fcgr2b, Fcgr4, Fcgr3, Fcgr1, Fcgrt, Ly6c1, Ly6c2, Ccr2, Cd36, Ly86. However, as the largest immune cell population in the mammalian testis, testicular macrophages (TMs) which expressed a large scale of macrophage specific markers will confuse the clustering of monocytes/macrophages. As TMs had been identified as M2-like macrophages^{17,18}, we successfully separate these cells from exogenic monocytes/macrophages by a M2 macrophages specific marker---CD163 (**see Fig. 1a and Supplementary Fig. S1a for details**).

Since the scRNA-Seq data is based on multiple subjects, it will be helpful to show t-SNE colored by individuals. Similarly, the proportions of cell types by individual will be helpful.

Reply: Thank you for your valuable comments. The t-SNE diagram of testicular cell clusters from ZIKV-infected or control mice were shown individually and distinguished by different colors (**see Fig. 1a for details**). The proportion and number of different cell clusters in ZIKV-infected or control mice testes were described in **lines 82-97** (**see Fig. 1b and Supplementary Fig. S1b for details**). And we found that the number of spermatogenic cells decreased significantly in ZIKV-infected testes, whereas the number of immune cells increased significantly.

- Fig 3C shows that S100A4 is expressed in the T cell cluster as well. It will be helpful if the authors can comment on that.

Reply: S100A4 gene was expressed in a variety of cells, including T cells, and had been proved to regulate the proliferation and migration of T cells, S100A4 deficiency leads to an inefficient integrin-dependent thymocyte migration and weaken the proliferation ability of mouse T cells in vitro¹⁹⁻²¹.

S100A4 transcriptome was really detected in T cells in our single cell RNA-Seq data. However, in our previous study, the expression of S100A4 protein in T cells is significantly lower than that in monocyte/macrophage⁶. We had analyzed the origin of S100A4+ cells by flow cytometry, by incubating testicular cells from ZIKV-infected A6 mice at 14dpi with antibodies against S100A4 and various other cellular markers including SOX9 (Sertoli cells), α -SMA (myoid epithelial cells), DDX4 (spermatogenic cells), CD11b (monocyte/macrophage), CD4 (lymphocyte) or CD8 (lymphocyte). Few S100A4+ cells expressed each of the

above cellular markers except for CD11b (See the figure below for details, a-f), for which approximately 80% of S100A4+ cells were positive (See the figure below for details, a). Therefore, we believe the high expression of S100A4 in monocyte/macrophage on both transcriptome and protein level was more persuasive, and the expression of S100A4 transcriptome in T cells still need more research.

- Line 67: most of them were expressed by monocytes or macrophages in clusters 0, 5, 7, and 22 – Cluster 5 is primarily composed of Fibroblasts. Are fibroblasts also involved in the complement activation?

Reply: Thank you for your comments. We are recently considering further research on this issue. Fibroblasts had been reported to synthesize a variety of complement system related molecules²²⁻²⁵ including C3 (see Fig. 3c, d for details). Therefore, these cells may promote the activation of complement system. Investigation on the expression of C3 protein in fibroblasts is ongoing in our lab. And, we also noticed that the number of fibroblasts in the testicular tissue of control mice was very rare, suggesting that the fibroblasts in the ZIKV-infected testes might be transformed from other cells (see Supplementary Fig. S1b for details).

Minor concerns:

- Panel 1c uses a different t-SNE compared to d-i.

Reply: We are sorry for this confusing panel and thank you for your reminding. We have modified the corresponding panels in this manuscript and shown the expression of DDX4 in different spermatogenic cells individually (see Fig. 2 for details).

- Providing the results of differential gene expression for Fig 1k-l with p-values as Supplementary Table.

Reply: Thank you for your suggestion. The P values of the genes of interest in testes from ZIKV-infected macaca leonine at 60dpi were shown in the table below and were provided as Supplementary Table 1 in this manuscript.

Gene name	P Value
Masp2	6.44E-07
C1s	0.001146
Cfp	0.276223
Masp1	0.634024
Cfi	0.704472
C1r	0.785686
C3	0.836768
Cd68	1.19E-06
Itgam/Cd11b	0.027609
Cd33	0.026016
Cd163	0.005772
S100a4	1
Mst1r	3.26E-05
Mst1	0.000668
Macir	0.183525
Ddx4	0.970474
Sycp1	0.400584

Vim	0.151591
-----	----------

- It has been mentioned that the ggplot2 package was used for image processing, base recognition and KEGG enrichment analysis. This needs to be confirmed.

Reply: Thank you for pointing this out. Image processing and base recognition were conducted by ggplot2 R package, and KEGG enrichment analysis were conducted by clusterProfiler R package. The detailed method of single-cell RNA sequencing is supplemented in Materials and Methods section in lines 613-688.

- The processed data should be uploaded to a public repository such as NCBI GEO.

Reply: Thank you for your advice. We've uploaded our processed data to NCBI GEO and all data in this manuscript are available from the corresponding author upon reasonable request.

References

1. Foy, B. D. et al. Probable non-vector-borne transmission of Zika virus, Colorado, USA. *Emerg. Infect. Dis.* **17**, 880-882 (2011).
2. Torres, J. R., Martinez, N. & Moros, Z. Microhematospermia in acute Zika virus infection. *Int. J. Infect. Dis.* **51**, 127 (2016).
3. Kurscheidt, F. A. et al. Persistence and clinical relevance of Zika virus in the male genital tract. *Nat. Rev. Urol.* **16**, 211-230 (2019).
4. Garcia-Bujalance, S. et al. Persistence and infectivity of Zika virus in semen after returning from endemic areas: Report of 5 cases. *J. Clin. Virol.* **96**, 110-115 (2017).
5. Atkinson, B. et al. Presence and Persistence of Zika Virus RNA in Semen, United Kingdom, 2016. *Emerg. Infect. Dis.* **23**, 611-615 (2017).
6. Yang, W. et al. S100A4+ macrophages facilitate zika virus invasion and persistence in the seminiferous tubules via interferon-gamma mediation. *PLoS Pathog.* **16**, e1009019 (2020).
7. Matusali, G. et al. Zika virus infects human testicular tissue and germ cells. *J. Clin. Invest.* **128**, 4697-4710 (2018).
8. Peregrine, J. et al. Zika Virus Infection, Reproductive Organ Targeting, and Semen Transmission in the Male Olive Baboon. *J. Virol.* **94**, (2019).
9. Zhao, L. et al. Single-cell analysis of developing and azoospermia human testicles reveals central role of Sertoli cells. *Nat. Commun.* **11**, 5683 (2020).
10. Jung, M. et al. Unified single-cell analysis of testis gene regulation and pathology in five mouse strains. *ELife.* **8**, (2019).
11. Robinson, C. L. et al. Male germ cells support long-term propagation of Zika virus. *Nat. Commun.* **9**, 2090 (2018).
12. Sheng, Z. Y. et al. Sertoli Cells Are Susceptible to ZIKV Infection in Mouse Testis. *Front Cell Infect Microbiol.* **7**, 272 (2017).
13. Stein, U. et al. Intervening in beta-catenin signaling by sulindac inhibits

- S100A4-dependent colon cancer metastasis. *Neoplasia*. **13**, 131-144 (2011).
14. Strutz, F. et al. Role of basic fibroblast growth factor-2 in epithelial-mesenchymal transformation. *Kidney Int.* **61**, 1714-1728 (2002).
 15. Okada, H., Danoff, T. M., Kalluri, R. & Neilson, E. G. Early role of Fsp1 in epithelial-mesenchymal transformation. *Am J Physiol.* **273**, F563-F574 (1997).
 16. Dahlmann, M., Kobelt, D., Walther, W., Mudduluru, G. & Stein, U. S100A4 in Cancer Metastasis: Wnt Signaling-Driven Interventions for Metastasis Restriction. *Cancers (Basel)*. **8**, (2016).
 17. Jaiswal, M. K. et al. Vacuolar-ATPase isoform a2 regulates macrophages and cytokine profile necessary for normal spermatogenesis in testis. *J Leukoc Biol.* **96**, 337-347 (2014).
 18. Meinhardt, A., Wang, M., Schulz, C. & Bhushan, S. Microenvironmental signals govern the cellular identity of testicular macrophages. *J Leukoc Biol.* **104**, 757-766 (2018).
 19. Bouchard, A. et al. Hippo Signal Transduction Mechanisms in T Cell Immunity. *Immune Netw.* **20**, e36 (2020).
 20. Salojin, K. V. et al. Genetic deletion of Mst1 alters T cell function and protects against autoimmunity. *PLoS One.* **9**, e98151 (2014).
 21. Cheng, J. et al. The Role of Mst1 in Lymphocyte Homeostasis and Function. *Front Immunol.* **9**, 149 (2018).
 22. Le Fournis, C., Jeanneau, C., Roumani, S., Giraud, T. & About, I. Pulp Fibroblast Contribution to the Local Control of Pulp Inflammation via Complement Activation. *J Endod.* **46**, S26-S32 (2020).
 23. Friscic, J. et al. The complement system drives local inflammatory tissue priming by metabolic reprogramming of synovial fibroblasts. *Immunity.* **54**, 1002-1021 (2021).
 24. Huang, Z., Feng, Y., Zhu, X., Wang, L. & Lu, W. MK801 regulates the expression of key osteoarthritis factors in osteoarthritis synovial fibroblasts through complement C5. *Res. Vet. Sci.* **136**, 377-384 (2021).
 25. McNearney, T., Ballard, L., Seya, T. & Atkinson, J. P. Membrane cofactor protein of complement is present on human fibroblast, epithelial, and endothelial cells. *J. Clin. Invest.* **84**, 538-545 (1989).

Reviewer comments, second round review

Reviewer #2 (Remarks to the Author):

This revised manuscript has addressed many of the concerns raised in the previous submission. There are, however, several issues that should be tackled for this paper to be an important contribution to the Zika literature. These are:

1. The language still needs to be tightened. Readers should not have to carefully study the previous publication to understand the context of this study. Line 68, "...but found out these were not the case" therefore, lacks clarity. Perhaps the authors would consider a one line summary of their previous findings, such as, "However, contrary to our exception, we found that CD8 T cells could not access the intraluminal space and hence interact with S100A4+ macrophages." Similar changes would also be helpful in many places throughout the manuscript.
2. Line 100. What is meant by substantial expression of ZIKV RNA?
3. Figures 1c, 3c and 3f. What units are these y-axis in?
4. Figure 1d. Please label the extreme right of the x-axis to indicate what those units represent.

Reviewer #3 (Remarks to the Author):

A few minor comments:

- The methods for transcriptomic analysis are still missing details. Lines 681-693. For example, the release for genome reference used, tool used for QC and what parameters, version number for packages and references for tools/packages
- Line 88-89 states that TM were present in both groups while Fig S1b states that they were absent in infected group.
- Fig 1b should be plotted showing proportion per animal and not pooled for the entire group
- Line 693 ggplot2 is not used for image processing and base recognition. This should be corrected. Did the authors mean bcl2fastq?
- Line 728 states that 25K cells passed QC while line 84-84 state it is 11014 cells in control testes and 11974 cells in ZIKV-infected testes

REVIEWER COMMENTS

Reviewer #2 (Remarks to the Author):

This revised manuscript has addressed many of the concerns raised in the previous submission. There are, however, several issues that should be tackled for this paper to be an important contribution to the Zika literature.

Reply: Thank you very much for your encouragement. We have replied to your questions and revised the manuscript. We hope that the manuscript can meet your requirements by these revision.

These are:

1. The language still needs to be tightened. Readers should not have to carefully study the previous publication to understand the context of this study. Line 68, "...but found out these were not the case" therefore, lacks clarity. Perhaps the authors would consider a one line summary of their previous findings, such as, "However, contrary to our exception, we found that CD8 T cells could not access the intraluminal space and hence interact with S100A4+ macrophages." Similar changes would also be helpful in many places throughout the manuscript.

Reply: Thank you for your suggestions. We've carefully revised the description about our previous study according to your comment. Please see lines 65-70 for details. Additionally, we asked American Journal Experts (AJE) to improve the English language according to your suggestion and the revised manuscript has also been reformatted. Hope all the revisions can address your concern.

2. Line 100. What is meant by substantial expression of ZIKV RNA?

Reply: By using substantial, we intended to emphasize that ZIKV RNA was detected in most cells. As this word may cause confusion, the description in revised manuscript is replaced by the statement of "In ZIKV-infected testes, ZIKV RNA was detected in most cells excluding Leydig cells and fibroblasts" (lines 101-102).

3. Figures 1c, 3c and 3f. What units are these y-axis in?

Reply: We are sorry for the mistake. The units of the y-axis in these figures are UMI count. UMI is the abbreviation of unique molecular identifier. For scRNA-seq, library preparation was performed for each cell based on the UMI protocol. After sequencing, gene expression was quantified by counting the UMI number of each gene, so the unit of expression level is the UMI count. The units had been added in Figures 1c, 3c, 3f and other relevant figures (Figures 2 and 5, Supplementary Figures S1, S5 and S6).

4. Figure 1d. Please label the extreme right of the x-axis to indicate what those units represent.

Reply: Thank you for your suggestion. The extreme right of Figure 1d shows the contribution of each pathway in the signaling patterns, and the label of the x-axis was added accordingly (Fig. 1d).

Reviewer #3 (Remarks to the Author):

A few minor comments:

- The methods for transcriptomic analysis are still missing details. Lines 681-693. For example, the release for genome reference used, tool used for QC and what parameters, version number for packages and references for tools/packages

Reply: Thank you for the comments. We've added more details of the transcriptomic analysis as your suggestion, including QC tool and its parameters, versions and references of all tools/packages, and release for genome reference. Please see lines 689-702 for details.

- Line 88-89 states that TM were present in both groups while Fig S1b states that they were absent in infected group.

Reply: We are very sorry for this error. We've deleted the misleading description of TM. The revised description is "Of them, Leydig cells, spermatogenic cells, Sertoli cells (suspected) and fibroblasts were testicular resident cells" (lines 90-91).

- Fig 1b should be plotted showing proportion per animal and not pooled for the entire group

Reply: Thank you for pointing this out. In this study, we used a mixed sample of three mice in each group for scRNA-seq, therefore, we could only show the percentage of each cell cluster in the entire group.

- Line 693 ggplot2 is not used for image processing and base recognition. This should be corrected. Did the authors mean bcl2fastq?

Reply: In our study, the ggplot2 was used to draw bar charts, pie charts and violin charts. However, the description may be improper, which was revised to be "A ggplot2 R package v3.4.0 is used to draw bar charts, pie charts and violin charts" (lines 701-702).

- Line 728 states that 25K cells passed QC while line 84-84 state it is 11014 cells in control testes and 11974 cells in ZIKV-infected testes

Reply: We are sorry for the mistake. The number of cells that passed QC was 22,988, and it was corrected in revised manuscript (lines 737 and 742).

Response to Reviewer comments

Reviewer #2 (Remarks to the Author):

This revised manuscript has addressed many of the concerns raised in the previous submission. There are, however, several issues that should be tackled for this paper to be an important contribution to the Zika literature.

Reply: Thank you very much for your encouragement. We have replied to your questions and revised the manuscript. We hope that the manuscript can meet your requirements by these revision.

These are:

1. The language still needs to be tightened. Readers should not have to carefully study the previous publication to understand the context of this study. Line 68, "...but found out these were not the case" therefore, lacks clarity. Perhaps the authors would consider a one line summary of their previous findings, such as, "However, contrary to our exception, we found that CD8 T cells could not access the intraluminal space and hence interact with S100A4+ macrophages." Similar changes would also be helpful in many places throughout the manuscript.

Reply: Thank you for your suggestions. We've carefully revised the description about our previous study according to your comment. Please see lines 65-70 for details. Additionally, we asked American Journal Experts (AJE) to improve the English language according to your suggestion and the revised manuscript has also been reformatted. Hope all the revisions can address your concern.

2. Line 100. What is meant by substantial expression of ZIKV RNA?

Reply: By using substantial, we intended to emphasize that ZIKV RNA was detected in most cells. As this word may cause confusion, the description in revised manuscript is replaced by the statement of "In ZIKV-infected testes, ZIKV RNA was detected in most cells excluding Leydig cells and fibroblasts" (lines 101-102).

3. Figures 1c, 3c and 3f. What units are these y-axis in?

Reply: *We are sorry for the mistake. The units of the y-axis in these figures are UMI count. UMI is the abbreviation of unique molecular identifier. For scRNA-seq, library preparation was performed for each cell based on the UMI protocol. After sequencing, gene expression was quantified by counting the UMI number of each gene, so the unit of expression level is the UMI count. The units had been added in Figures 1c, 3c, 3f and other relevant figures (Figures 2 and 5, Supplementary Figures S1, S5 and S6).*

4. Figure 1d. Please label the extreme right of the x-axis to indicate what those units represent.

Reply: *Thank you for your suggestion. The extreme right of Figure 1d shows the contribution of each pathway in the signaling patterns, and the label of the x-axis was added accordingly (Fig. 1d).*

Reviewer #3 (Remarks to the Author):

A few minor comments:

- The methods for transcriptomic analysis are still missing details. Lines 681-693. For example, the release for genome reference used, tool used for QC and what parameters, version number for packages and references for tools/packages

Reply: *Thank you for the comments. We've added more details of the transcriptomic analysis as your suggestion, including QC tool and its parameters, versions and references of all tools/packages, and release for genome reference. Please see lines 689-702 for details.*

- Line 88-89 states that TM were present in both groups while Fig S1b states that they were absent in infected group.

Reply: *We are very sorry for this error. We've deleted the misleading description of TM. The revised description is "Of them, Leydig cells, spermatogenic cells, Sertoli cells (suspected) and fibroblasts were testicular resident cells" (lines 90-91).*

- Fig 1b should be plotted showing proportion per animal and not pooled for the entire group

Reply: *Thank you for pointing this out. In this study, we used a mixed sample of three mice in each group for scRNA-seq, therefore, we could only show the percentage of each cell cluster in the entire group.*

- Line 693 ggplot2 is not used for image processing and base recognition. This should be corrected. Did the authors mean bcl2fastq?

Reply: *In our study, the ggplot2 was used to draw bar charts, pie charts and violin charts. However, the description may be improper, which was revised to be "A ggplot2 R package v3.4.0 is used to draw bar charts, pie charts and violin charts" (lines 701-702).*

- Line 728 states that 25K cells passed QC while line 84-84 state it is 11014 cells in control testes and 11974 cells in ZIKV-infected testes

Reply: *We are sorry for the mistake. The number of cells that passed QC was 22,988, and it was corrected in revised manuscript (lines 737 and 742).*